# Circadian rhythm profiles derived from accelerometer measures of the sleep-wake cycle in two cohort studies

Sam Vidil [1], Ian Meneghel Danilevicz [1], Aline Dugravot [1], Aurore Fayosse [1], Benjamin Landré[1], Vincent van Hees[2], Mathilde Chen [3,4], Archana Singh-Manoux[1,5] & Séverine Sabia [1,5] ✉

Accelerometers allow objective measures of dimensions (rest-activity rhythm (RAR), daytime activity, sleep, and chronotype) of the bio-behavioural manifestation of circadian rhythm (CR) using multiple metrics in large-scale studies. These dimensions are rarely examined together due to methodological challenges of using correlated data. To address this challenge, we propose a two-step approach consisting of data reduction of CR metrics using principal component analyses, followed by k-means clustering to identify groups of individuals with a similar profile using data from the Whitehall II (N = 3,991, mean age=69.4years) and UK Biobank (N = 54,995, mean age=67.5years) cohort studies. Our analyses identified nine CR clusters: two presented extreme (most robust/poorest) RAR and (highest/lowest) daytime activity, two robust RAR with opposite sleep profiles (longer and efficient/shorter and fragmented), one high-intensity physical activity, and four poor RAR (one characterised by late chronotype, two by low activity but opposite sleep profiles, and one by restless (agitated) sleep). The participants in these nine clusters differed on sociodemographic, behavioural and health-related factors. Findings were similar in these two independent cohort studies, highlighting the validity of our approach. Most previous studies have used only the RAR dimension of circadian rhythm, and here we show that this might be an oversimplification as demonstrated by nine clusters characterised by combinations of RAR, daytime activity, sleep, and chronotype. Our innovative approach demonstrates feasibility of using all dimensions to study the impact of circadian rhythm dysregulation on health.

Several key human body functions and biological processes are regulated over a 24 h cycle by an internal biological clock[1]. Regulation of this rhythm is progressively disrupted with ageing, as manifested by shifts in biological processes, and a range of behaviours such as the sleep-wake cycle[2]. There is also emerging evidence of a link between

dysregulation in an organism's innate timing device—the circadian clock[3], and its bio-behavioural manifestation—circadian rhythm[2], and a range of chronic diseases. Along with studies on the molecular organisation of the circadian clock[3], there is increasing research interest in its bio-behavioural manifestation; recent advances in accelerometry

[1]Université Paris Cité, Inserm U1153, Center for Research in Epidemiology and Statistics (CRESS), Epidemiology of Ageing and Neurodegenerative Diseases (EpiAgeing), Paris, France. [2]Accelting, Almere, The Netherlands. [3]CIRAD, UMR PHIM, Montpellier, France. [4]PHIM, CIRAD, INRAE, Institut Agro, IRD Université de Montpellier, Montpellier, France. [5]Faculty of Brain Sciences, University College London, London, UK. ✉e-mail: s.sabia@ucl.ac.uk

allow scalable, objective measures of this behavioural circadian rhythm in large-scale studies[4,5].

Multiple metrics can be derived from 24 h accelerometer data to reflect four key behavioural dimensions of circadian rhythm—'rest-activity rhythm' (RAR) reflecting circadian rhythmicity in a free-living setting, 'daytime activity' composed of physical activity (PA) and sedentary behaviour (SB) over the waking period, 'sleep' to measure the quality and quantity of sleep during the sleep period, and 'chronotype' to measure wakefulness and sleep timing. Whether dysregulation is uniform across these circadian rhythm dimensions in individuals remains unclear. Most studies on accelerometer-based measures of circadian rhythm focus on RAR and chronotype, either ignoring other dimensions[6], or using only one marker of PA and/or sleep[7-12]. Few studies have examined all four dimensions (RAR, daytime activity, sleep, and chronotype)[13-15] and have either considered these dimensions in separate models[13], or in a mutually adjusted model[14], or derived a large number of composite scores[15], that do not allow individual patterns of dysregulations in the four dimensions to be elucidated.

The quantity of data generated by accelerometers, with multiple metrics for each dimension, and their correlated nature present analytic challenges. The aim of this study was to consider all four dimensions (RAR, daytime activity, sleep, and chronotype) and their metrics to identify real-life human clusters of circadian rhythm in older adults. To address concerns of the correlation between various metrics, we used a two-step procedure consisting of data reduction and then cluster analysis. To ensure construct validity of the method, analyses were undertaken in the Whitehall II (WII) and in the UK Biobank (UKB) cohort studies. A secondary objective was to determine the socio-demographic, behavioural, and health-related factors correlates of circadian rhythm clusters identified in our study.

## Results

A total of 3,991 (mean age = 69.4 (standard deviation (SD) = 5.7, range = 60-83) years; 26% women) and 54,995 (mean age = 67.5 (SD = 4.2, range = 60-79) years; 54% women) participants from WII and UKB, respectively, were included in the clustering analyses (flowcharts in Figs. S1 and S2). The correlations (Fig. 1) between the 36 accelerometer-based metrics (described in Table S1) were mostly moderate (0.40 to 0.59) to high (≥0.60) for metrics within the same dimension of circadian rhythm—RAR, daytime activity, sleep, and chronotype. RAR metrics were also highly correlated with most daytime activity metrics (e.g. absolute range of correlations of cosinor mesor with 13 out of 15 daytime activity metrics: 0.53–0.86 in WII, 0.41–0.83 in UKB), moderately correlated with some sleep metrics (absolute range of correlations of relative amplitude with 6 out of 10 sleep metrics: 0.27–0.57 in WII, 0.17–0.76 in UKB), and had a weak correlation with chronotype metrics (absolute range of correlations for the 5 metrics: 0.11–0.29 in WII, 0.11–0.31 in UKB). The correlations between metrics of daytime activity, sleep, and chronotype were weaker (<0.40).

To justify use of 36 rather than commonly used 10 circadian rhythm metrics (including interdaily stability, intradaily variability, relative amplitude, cosinor mesor, cosinor amplitude, cosinor acro-time, $M_{10}$ and $L_5$ timing and mean acceleration, excluding most daytime activity and sleep metrics) we compared the predictive performance for mortality of the 10 metrics and all 36 metrics. This was done by first using principal components analysis (PCA) in each set of metrics (10 and 36 metrics). A total of three principal components were retained in analysis on 10 metrics (Tables S2 and S3) that explained 80.4% and 77.6% of the variability in WII and UKB datasets, respectively. For 36 metrics, eight principal components were retained that explained 86.5% and 85.4% of the variability in WII and UKB datasets, respectively (Table S4). Over median 11.0 and 8.0 years of follow-up, a total of 633 and 3256 participants died in WII and UKB,

respectively. The predictive performance of the model with 36 metrics (C-index = 0.675, 95% confidence interval (CI) = 0.653–0.696 in WII; and C-index = 0.652, 95% CI = 0.643–0.662 in UKB) was higher than the model with 10 metrics (C-index = 0.651, 95% CI = 0.629–0.672 in WII; and C-index=0.622, 95% CI = 0.612–0.632 in UKB), p for difference <0.001 for both cohorts. Subsequent analyses were therefore based on 36 metrics.

### Identification of circadian rhythm clusters

The first step in the analysis (procedure shown in Fig. S3), principal component analyses (PCA) for data reduction of the 36 metrics, yielded the eight principal components (described above, Tables S4–S6). K-means cluster analysis on these principal components found nine clusters to be the optimal representation of circadian rhythm metrics in participants in both cohort studies. This decision was based on examination of statistical criteria (Supplementary Figs. S4 and S5) and interpretability of clusters.

The standardised and raw mean and SD of the 36 metrics for participants in each cluster are shown in Fig. 2 and Table S7 (WII) and Fig. 3 and Table S8 (UKB). A plot of the acceleration signal for a medoid participant (most central individual) in each cluster is shown in Supplementary Figs. S6 (WII) and S7 (UKB). Ranking the mean values by magnitude in each cluster (Tables S7 and S8) allowed their interpretation in relation to RAR, daytime activity, sleep, and chronotype.

Clusters were the same in both cohort studies, unless otherwise specified. Participants in cluster 1 (RAR++/PA++) had the most robust (++) RAR—characterised by highest amplitude, less fragmented and more stable rhythm —and active (++) pattern both in terms of light-intensity PA (LIPA) and moderate-to-vigorous PA (MVPA) while those in cluster 9 (RAR−−/PA−−) had the poorest (−−) RAR and least (−−) active pattern. In UKB, cluster 9 (RAR−−/PA−/Chronotype−) was also characterised by delayed chronotype. Participants in clusters 2 and 3 had robust (+) RAR but opposite sleep profiles with cluster 2 characterised by shorter sleep duration (in UKB), more innefficient and fragmented sleep (Sleep−) and cluster 3 by longer sleep duration, more efficient, and less fragmented sleep (Sleep+). In addition, in WII, cluster 2 (RAR+/LIPA+/Sleep−) was also characterised by more LIPA and in UKB by both more LIPA and MVPA (RAR+/PA+/Sleep−). In both WII and UKB, cluster 3 (RAR+/LIPA+/Sleep+) participants had daytime activity characterised by light-intensity PA (LIPA+). Participants in cluster 4 (MVPA++) had a RAR pattern close to the mean, but with higher relative amplitude that was characterised by higher values in metrics related to high-intensity PA, including MVPA features and intensity gradient (IG) slope.

Participants in clusters 5, 6, 7, and 8 had poor (−) RAR pattern. Cluster 5 (RAR−/Chronotype−−) had the most delayed (−−) chronotype with later sleep period and later activity during the day. Participants in clusters 6 (RAR−/PA−/Sleep+) and 7 (RAR−/PA−/Sleep−−) had both low daytime activity but differed in sleep. Participants in cluster 8 (RAR−/PA+/Restless sleep) showed restless (agitated) sleep where mean acceleration during sleep was the highest (>1 SD and 2 SD from the sample mean in WII and UKB, respectively). In WII, participants in cluster 8 (RAR−/PA+/Restless sleep) also presented shorter and less efficient sleep. In this cluster, relative amplitude was low and cosinor mesor high, resulting from both more daytime PA and more nighttime movements (measured by mean acceleration) compared to participants in other clusters.

Clusters 3 (RAR+/LIPA+/Sleep+) and 6 (RAR−/PA−/Sleep+) in both cohort studies were the largest (15.7/17.1% (WII/UKB) and 17.7/16.1%, respectively) and clusters 8 (RAR−/PA+/Restless sleep) and 9 (RAR−−/PA−−/(Chronotype−, in UKB)) the smallest (5.9/3.4% and 6.5/6.9%, respectively). The size of clusters was similar in WII and UKB, with the largest difference observed for cluster 4 (MVPA++) (9.5 and 13.0%). Data on subjective chronotype preference in UKB (Table S9) showed 8.2% of participants to report themselves as being 'Definitely an

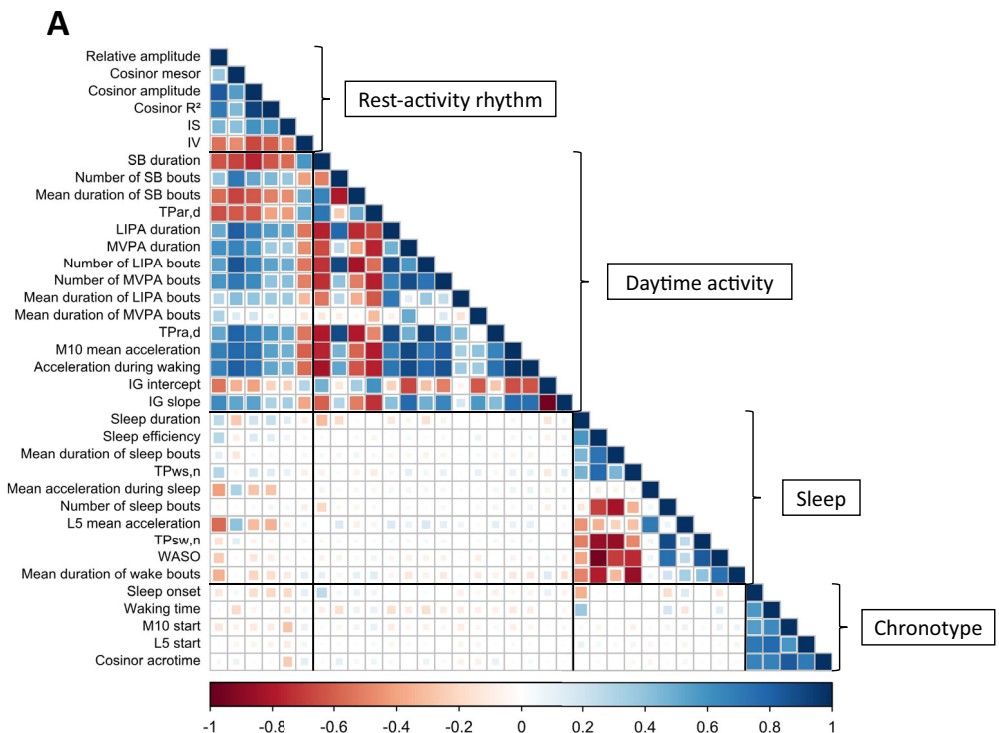

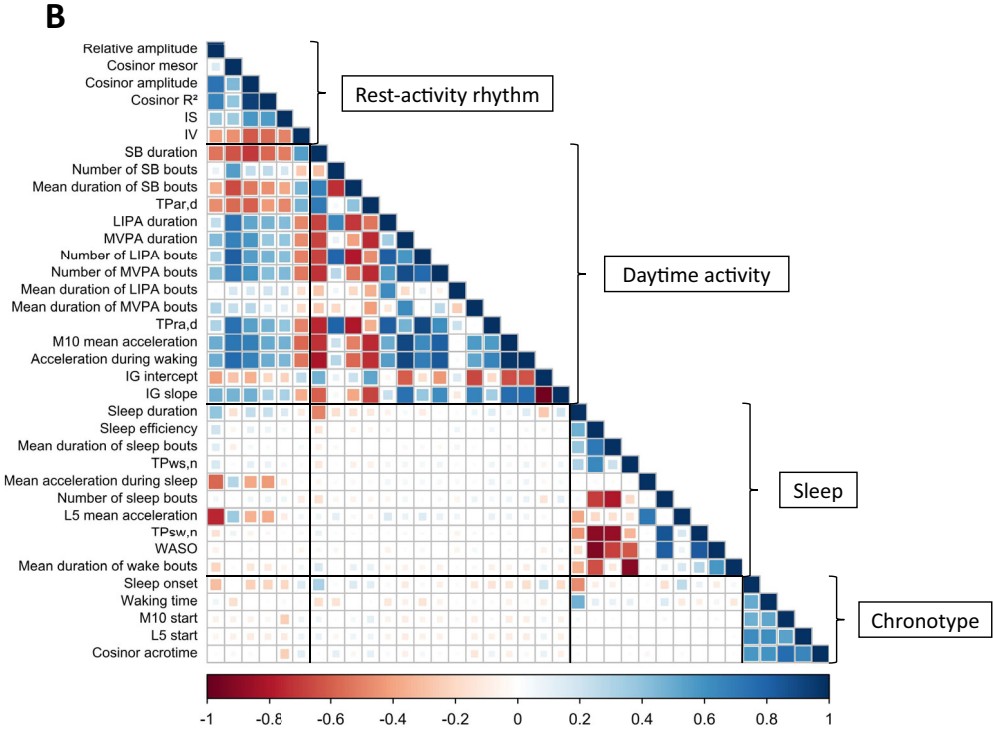

**Fig. 1 | Correlation matrix of 36 metrics reflecting circadian rhythm dimensions (rest-activity rhythm, daytime activity, sleep, and chronotype). a Whitehall II. b UK Biobank.** IG intensity gradient, $L_5$ least active 5-h period, LIPA light intensity physical activity, $M_{10}$ most active 10-h period, MVPA moderate to vigorous physical activity, SB sedentary behaviour, TPar,d transition probability from activity to rest during the day, TPra,d transition probability from rest to activity during the day, TPsw,n transition probability from sleep to wake during the night, TPws,n transition probability from wake to sleep during the night, WASO wake after sleep onset. A day is defined as the period between waking time to start the day and sleep onset, and a night as the period between sleep onset and waking for the day. Source data are provided as a Source Data file.

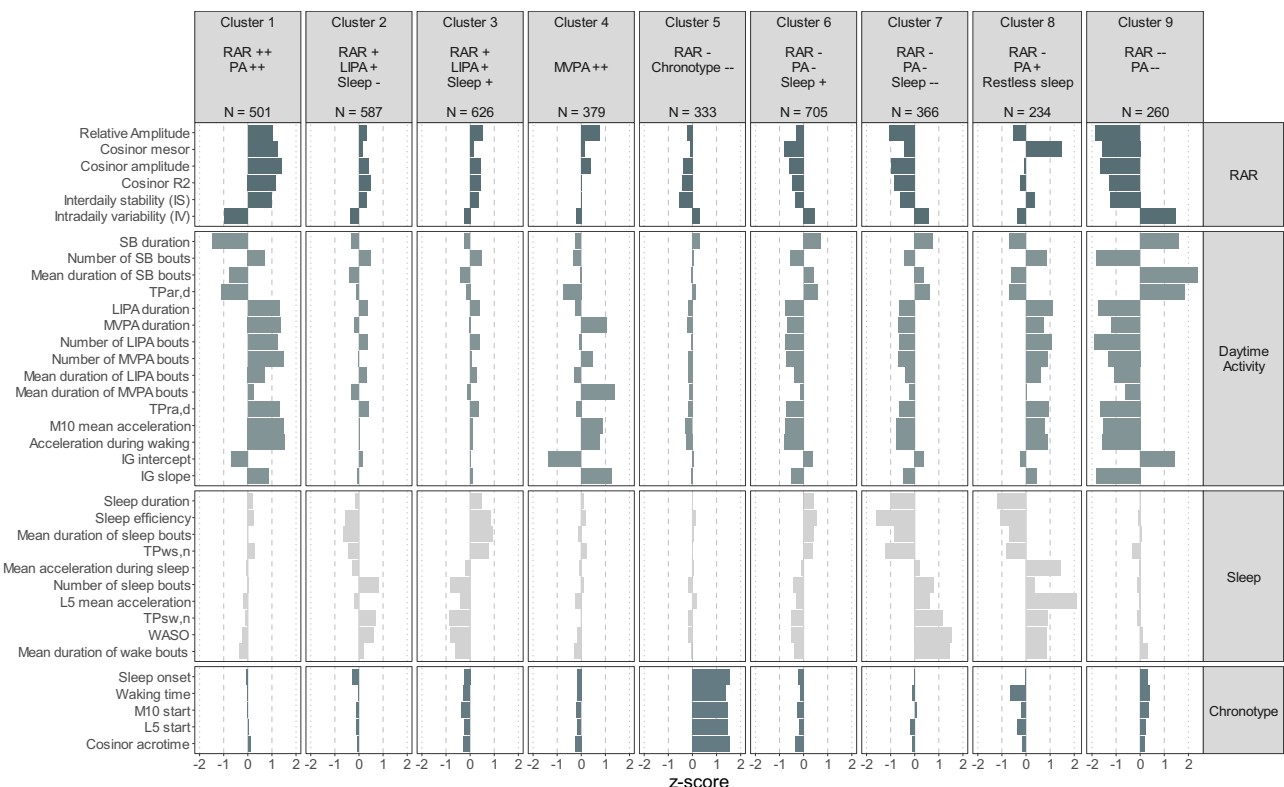

**Fig. 2 | Standardised mean scores on 36 metrics as a function of nine circadian rhythm clusters in the Whitehall II accelerometer sub-study.** IG intensity gradient, L5 least active 5-h period, LIPA light intensity physical activity, M10 most active 10-h period, MVPA moderate to vigorous physical activity, PA physical activity, RAR rest-activity rhythm, SB sedentary behaviour, TPar,d transition probability from activity to rest during the day, TPra,d transition probability from rest to activity during the day, TPsw,n transition probability from sleep to wake during the night, TPws,n transition probability from wake to sleep during the night, WASO wake after sleep onset. Results shown in the figure are mean standardised scores (z-scores) for each metric in each cluster. Source data are provided as a Source Data file.

evening person' and this was particularly the case for participants in clusters identified with delayed chronotype using accelerometer data −clusters 5 (RAR−/Chronotype−−) and 9 (RAR−−/PA−−/Chronotype−) (21.9% and 16.1%, respectively, compared to 6.0% in cluster 3 (RAR+/LIPA+/Sleep+), p < 0.001).

**Factors associated with circadian rhythm clusters**
Table 1 (WII) and Table 2 (UKB) show participants' characteristics in the nine circadian rhythm clusters among participants with complete data (N = 3968 in WII and N = 51,507 in UKB). Participants in UKB who worked shifts (N/Total N with available data=4441/34,467) were more likely to be in cluster 1 (RAR++/PA++), 2 (RAR+/PA+/Sleep−), 7 (RAR−/PA−/Sleep−−) and 9 (RAR−−/PA−−/Chronotype−), all p ≤ 0.01 for comparison with cluster 3 (RAR+/LIPA+/Sleep+), Table S10. Associations of sociodemographic, behavioural, and health-related factors (independent variables) with circadian rhythm clusters (dependent variable) were examined using multinomial regressions. These covariates were measured concurrently to accelerometer measures except, in UKB, for marital status, employment status, Townsend deprivation index, behavioural factors, body mass index (BMI), hyperlipidaemia, and central nervous system (CNS) medication, that were measured in 2006–2010 in average (SD) 5.7 (1.1) years before accelerometer measure).

In these analyses, cluster 3 (RAR+/LIPA+/Sleep+) was used as the reference due to its size (N = 619, 15.6%, in WII and N = 8811, 17.1%, in UKB), and a healthy profile with all mean values on all metrics being between −1 and +1 SD. These results (Table 3 for WII and Table 4 for UKB) show participants in clusters characterised by disrupted RAR and low PA, irrespective of sleep, to be older; these were clusters 6

(RAR−/PA−/Sleep+), 7 (RAR−/PA−/Sleep−−), and 9 (RAR−− /PA− −(/Chronotype− in UKB)). The reference cluster was more likely to be composed of women; all p < 0.015. Non-white participants were more likely to be part of clusters characterised by poor sleep (clusters 2, 7, and 8), late chronotype (cluster 5), and in WII also poorer RAR/ activity profile (cluster 9); all p < 0.01. Participants who were not married/cohabiting were more likely to be in the reference cluster 3 (RAR+/LIPA+/Sleep+) in both cohort studies; all p < 0.044; except for cluster 2 (RAR+/LIPA+/Sleep−) and 6 (RAR−/PA−/Sleep+) in WII where no difference were seen with cluster 3. Participants in employment were more likely to be in cluster 8 (RAR−/PA+/Restless sleep) and less likely to be in clusters 9 (RAR−−/PA−−(/Chronotype− in UKB)) and 5 (RAR−/Chronotype−−) in WII, and in cluster 4 (MVPA++) and 6 (RAR−/PA−/Sleep+) in UKB. WII participants living in more deprived areas were more likely to be in clusters characterised by disturbed sleep and late chronotype (clusters 2, 5, 7, and 8) and in UKB, they were more likely to be in the cluster with high-intensity PA (cluster 4) and all clusters characterised by poorer RAR (clusters 5 to 9).

Among behavioural factors, we found current smokers to be more likely to be in clusters 2 (RAR+/LIPA+/Sleep−), 5 (RAR−/Chronotype−−), 7 (RAR−/PA−/Sleep−−), and 9 (RAR−−/PA−−(/Chronotype− in UKB)); this was also the case for cluster 6 (RAR−/PA−/Sleep+) in WII and 8 (RAR−/PA+/Restless sleep) in UKB. Individuals who consumed less fruit and vegetables were more likely to be in clusters 7 (RAR−/PA−/Sleep−−) and 9 (RAR−−/PA−−(/Chronotype− in UKB)), and in UKB also in clusters 5 (RAR−/Chronotype−−) and 6 (RAR−/PA−/Sleep+). Exposure to light was higher in cluster 1 (RAR++/PA++) that had the most robust RAR and highest PA and lower in clusters characterised by poorer RAR and less PA (clusters 5, 6, 7, and 9); all p < 0.001.

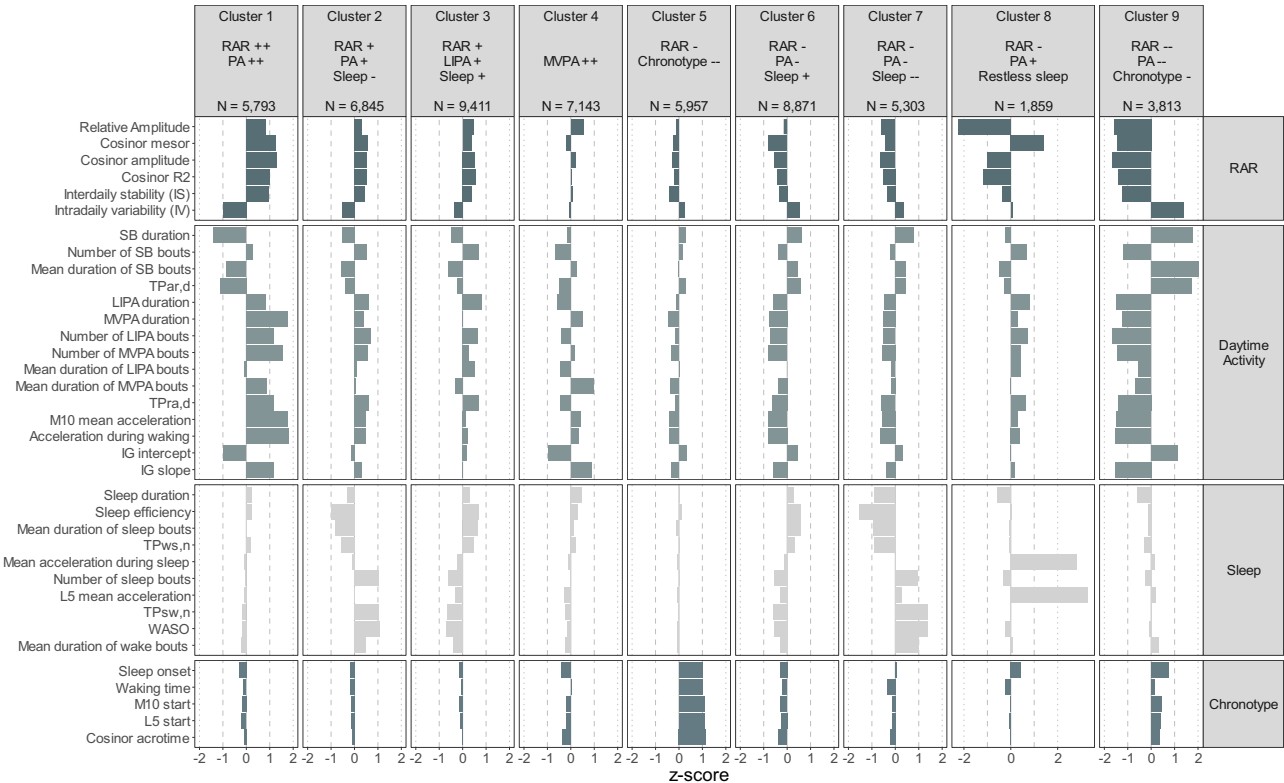

**Fig. 3 | Standardised mean scores on 36 metrics as a function of nine circadian rhythm clusters in the UK Biobank accelerometer sub-study.** IG intensity gradient, L5 least active 5-h period, LIPA light intensity physical activity, M10 most active 10-h period, MVPA moderate to vigorous physical activity, PA physical activity, RAR rest-activity rhythm, SB sedentary behaviour, TPar,d transition probability from activity to rest during the day, TPra,d transition probability from rest to activity during the day, TPsw,n transition probability from sleep to wake during the night, TPws,n transition probability from wake to sleep during the night, WASO wake after sleep onset. Results shown in the figure are mean standardised scores (z-scores) for each metric in each cluster. Source data are provided as a Source Data file.

Participants with higher BMI were less likely to be in clusters with higher PA (1 and 4), although the association was not statistically significant in WII (p = 0.29), and more likely to be in clusters with poorer RAR (clusters 5, 6, 7, 8, and 9), all p < 0.001 except for cluster 8 (RAR−/PA+/Restless sleep) in WII. Participants with diabetes were less likely to be in clusters characterised by MVPA (cluster 1 in UKB and cluster 4 in both studies), and more likely to be in the less active clusters in UKB (6, 7, and 9), while associations were in the same direction in WII although not statistically significant (p = 0.08, p = 0.31, p = 0.23, respectively). In UKB, participants with diabetes were also more likely to be in clusters 5 (RAR−/Chronotype−−) and 8 (RAR−/PA+/Restless sleep). No strong associations were found for hypertension and hyperlipidaemia, besides higher odds (1.26 (95% CI = 1.15–1.38)) of hypertension among participants in cluster 9 (RAR−−/PA−−) compared to cluster 3 (RAR+/LIPA+/Sleep+) in UKB. Participants with at least one chronic disease were more likely to be in cluster 5 (RAR−/Chronotype−−) and cluster 9 (RAR−−/PA−−(/Chronotype− in UKB)) in both studies, and in UKB also in clusters 6 (RAR−/PA−/Sleep+) and 7 (RAR−/PA−/Sleep−−). Use of CNS medication was less likely among participants in cluster 1 (RAR++/PA++) and more likely in clusters with poorer RAR (5 to 9).

## Discussion

The main premise of this study is that rest-activity rhythm (RAR) and chronotype dimensions alone may not comprehensively capture the bio-behavioural expression of circadian rhythm patterns in real-world settings. We identified circadian rhythm clusters in two population-based cohort studies using a method that allows clusters to be identified using a combination of metrics that reflect dimensions of the behavioural expression of circadian rhythm (36 metrics to reflect RAR, daytime activity, sleep, and chronotype). This approach is in contrast to studies that either use metrics related to a few dimensions or use dimensions of circadian rhythm independently of each other. The stability of clusters in our analysis was confirmed in two independent large-scale population-based cohort studies. The characteristics of individuals in these clusters differed as a function of socio-demographic, behavioural, and health-related factors, highlighting the validity of these clusters. Age, sex, ethnicity, cohabitation status, deprivation index, smoking, fruit and vegetable consumption, light exposure, BMI, diabetes, prevalence of chronic diseases, and CNS drugs were the factors that differed the most across the nine circadian rhythm clusters.

Circadian rhythm is altered at older ages[2] and in individuals with chronic diseases[2,3]. There is increasing interest in using accelerometer-based measures of RAR and/or chronotype to examine associations of circadian rhythm with adverse health outcomes[8,12,14]. These studies have generally not considered daytime activity or sleep to measure circadian rhythm, and cannot be seen as a comprehensive reflection of the sleep-wake cycle. Some studies[6,9,16] have included $M_{10}$ (mean acceleration during most active 10 h) and $L_5$ (mean acceleration during 5 least active hours) as broad indicators of PA and sleep, respectively, but this approach does not fully capture differences in activity intensity (for example LIPA vs MVPA) or sleep disruptions such as fragmentation (for example number of sleep bouts). Previous studies have shown both sleep[17,18] and PA[19], two key dimensions of the sleep-wake cycle, to be important for health. Results from our preliminary analysis show use of all 36 metrics to have better predictive performance for mortality than the prediction

**Table 1 | Characteristics of participants as a function of nine circadian rhythm clusters in the Whitehall II accelerometer sub-study**

| | Total study population | Cluster 1 RAR ++ PA ++ | Cluster 2 RAR + LIPA + Sleep– | Cluster 3 RAR + LIPA + Sleep + | Cluster 4 MVPA ++ | Cluster 5 RAR– Chronotype –– | Cluster 6 RAR – PA – Sleep + | Cluster 7 RAR – PA – Sleep –– | Cluster 8 RAR – PA + Restless sleep | Cluster 9 RAR –– PA–– |
|---|---|---|---|---|---|---|---|---|---|---|
| N (%) | 3968 | 495 (12.5) | 585 (14.7) | 619 (15.6) | 378 (9.5) | 332 (8.4) | 702 (17.7) | 363 (9.1) | 234 (5.9) | 260 (6.6) |
| **Socio-demographic factors** | | | | | | | | | | |
| Age (years), M (SD) | 69.4 (5.7) | 67.2 (4.6) | 69.3 (5.5) | 69.0 (5.6) | 66.9 (4.7) | 68.5 (5.4) | 71.1 (5.8) | 70.9 (6.1) | 69.0 (5.3) | 73.1 (5.7) |
| Sex, women | 1028 (25.9) | 151 (30.5) | 158 (27.0) | 206 (33.3) | 55 (14.6) | 86 (25.9) | 174 (24.8) | 66 (18.2) | 53 (22.6) | 79 (30.4) |
| Non-white ethnicity | 298 (7.5) | 19 (3.8) | 42 (7.2) | 25 (4.0) | 7 (1.9) | 34 (10.2) | 45 (6.4) | 58 (16.0) | 27 (11.5) | 41 (15.8) |
| Lower secondary school or less | 1647 (41.5) | 187 (37.8) | 279 (47.7) | 256 (41.4) | 144 (38.1) | 122 (36.7) | 273 (38.9) | 148 (40.8) | 102 (43.6) | 136 (52.3) |
| Not married/cohabiting | 1008 (25.4) | 110 (22.2) | 121 (20.7) | 131 (21.2) | 94 (24.9) | 122 (36.7) | 163 (23.2) | 101 (27.8) | 59 (25.2) | 107 (41.2) |
| In employment | 758 (19.1) | 91 (18.4) | 108 (18.5) | 126 (20.4) | 104 (27.5) | 57 (17.2) | 122 (17.4) | 66 (18.2) | 66 (28.2) | 18 (6.9) |
| Area index of multiple deprivation, M (SD) | 12.8 (9.9) | 11.5 (8.5) | 12.8 (9.2) | 11.5 (8.8) | 12.4 (9.5) | 15.2 (11.2) | 11.8 (9.4) | 15.0 (11.9) | 13.7 (10.8) | 14.2 (11.0) |
| **Behavioural factors** | | | | | | | | | | |
| Smoking | | | | | | | | | | |
| Never smokers | 1941 (48.9) | 251 (50.7) | 281 (48.0) | 322 (52.0) | 184 (48.7) | 141 (42.5) | 357 (50.9) | 176 (48.5) | 112 (47.9) | 117 (45.0) |
| Ex-smokers | 1898 (47.8) | 237 (47.9) | 282 (48.2) | 287 (46.4) | 188 (49.7) | 166 (50.0) | 324 (46.2) | 173 (47.7) | 118 (50.4) | 123 (47.3) |
| Smokers | 129 (3.3) | 7 (1.4) | 22 (3.8) | 10 (1.6) | 6 (1.6) | 25 (7.5) | 21 (3.0) | 14 (3.9) | 4 (1.7) | 20 (7.7) |
| Alcohol consumption | | | | | | | | | | |
| No consumption | 811 (20.4) | 79 (16.0) | 109 (18.6) | 113 (18.3) | 47 (12.4) | 82 (24.7) | 162 (23.1) | 86 (23.7) | 48 (20.5) | 85 (32.7) |
| 1–14 units/week | 2072 (52.2) | 256 (51.7) | 298 (50.9) | 360 (58.2) | 213 (56.3) | 157 (47.3) | 368 (52.4) | 177 (48.8) | 108 (46.2) | 135 (51.9) |
| > 14 units/week | 1085 (27.4) | 160 (32.3) | 178 (30.4) | 146 (23.6) | 118 (31.2) | 93 (28.0) | 172 (24.5) | 100 (27.5) | 78 (33.3) | 40 (15.4) |
| Fruits and vegetables, <twice a day | 1672 (42.1) | 173 (34.9) | 253 (43.2) | 208 (33.6) | 140 (37.0) | 151 (45.5) | 285 (40.6) | 198 (54.5) | 117 (50.0) | 147 (56.5) |
| Daily % >1000 lux, M (SD) | 13.6 (12.4) | 22.8 (15.6) | 15.5 (12.5) | 15.3 (11.7) | 12.9 (11.6) | 9.4 (8.9) | 9.7 (9.4) | 8.9 (8.2) | 18.5 (14.7) | 6.3 (7.0) |
| **Health-related factors** | | | | | | | | | | |
| BMI (kg/m²), M (SD) | 26.6 (4.3) | 24.9 (3.7) | 26.9 (3.9) | 25.7 (4.0) | 25.5 (3.5) | 26.9 (4.2) | 27.3 (4.2) | 28.2 (4.8) | 26.1 (4.3) | 29.0 (5.6) |
| Diabetes | 502 (12.7) | 29 (5.9) | 75 (12.8) | 59 (9.5) | 19 (5.0) | 41 (12.3) | 122 (17.4) | 73 (20.1) | 21 (9.0) | 63 (24.2) |
| Hypertension | 2351 (59.3) | 216 (43.6) | 372 (63.6) | 333 (53.8) | 186 (49.2) | 179 (53.9) | 484 (68.9) | 243 (66.9) | 133 (56.8) | 205 (78.8) |
| Hyperlipidaemia | 2014 (50.8) | 199 (40.2) | 309 (52.8) | 301 (48.6) | 162 (42.9) | 163 (49.1) | 393 (56.0) | 208 (57.3) | 119 (50.9) | 160 (61.5) |
| Prevalence of ≥1 chronic diseases[a] | 1590 (40.1) | 149 (30.1) | 229 (39.1) | 228 (36.8) | 110 (29.1) | 148 (44.6) | 304 (43.3) | 170 (46.8) | 86 (36.8) | 166 (63.8) |
| CNS medication | 256 (6.5) | 15 (3.0) | 32 (5.5) | 34 (5.5) | 10 (2.6) | 30 (9.0) | 39 (5.6) | 34 (9.4) | 23 (9.8) | 39 (15.0) |

*ADL* activity of daily living, *BMI* body mass index, *CNS* central nervous system, *LIPA* light intensity physical activity, *M* mean, *MVPA* moderate to vigorous physical activity, *PA* physical activity, *RAR* rest-activity rhythm, *SD* standard deviation.
[a]From a list composed of coronary heart disease, stroke, Parkinson's disease, chronic obstructive pulmonary disease, heart failure, depression, other mental disorders, cancer, liver disease, arthritis. Value are N (%), unless otherwise stated. Results rounded to one decimal place.

model with 10 commonly used metrics (RAR, chronotype, $M_{10}$ and $L_5$). Furthermore, eight of the nine circadian rhythm profiles we identified were characterised by RAR but also presented differences in the other dimensions. These findings suggest that focusing on RAR and/or chronotype alone may introduce misclassification bias in studies of sleep-wake cycles and health.

A thorough search of the relevant literature did not identify previous accelerometer-based studies on identification of distinct circadian rhythm profiles using metrics of RAR, daytime activity, sleep, and chronotype. Some studies have identified profiles based either on hourly activity levels[20–22], RAR metrics[7], daytime activity[23,24], or sleep[25,26] metrics. Other studies combined daytime activity and sleep metrics and found between three[27] and five[28] clusters. Another study combined RAR, sleep, and chronotype metrics and identified three profiles that differed primarily on sleep characteristics[29]. One notable study used data reduction on 28 metrics of the four circadian rhythm dimensions to derive 13 component scores[15] that essentially reflect the correlational structure of the circadian rhythm metrics. These results are comparable to the first step in our analysis—the PCA results where the first three components reflected RAR and daytime activity, sleep, and chronotype, while the others principal components more complex combinations of the metrics.

We added a second step by undertaking cluster analyses on the components from the PCA. The advantage of cluster analysis is that it allows individuals to be grouped in clusters that have high within-group similarity and between-group differences, while also accounting for the interrelated structure of the data. Our findings show how measures of RAR, daytime activity, sleep, and chronotype are distributed in the population. For example, clusters 5 (RAR−/Late chronotype), 6 (RAR−/PA−/Sleep+), and 7 (RAR−/PA−/Sleep−−) all have a similar pattern of poorer RAR but differed in the intensity and duration of daytime activity, chronotype, and sleep characteristics. Whether the longitudinal associations of these clusters with health outcomes differ needs to be examined in future studies.

**Table 2 | Characteristics of participants as a function of nine circadian rhythm clusters in the UK Biobank accelerometer sub-study**

| | Total study population | Cluster 1 RAR ++ PA ++ | Cluster 2 RAR + PA + Sleep− | Cluster 3 RAR + LIPA + Sleep + | Cluster 4 MVPA ++ | Cluster 5 RAR − Chronotype −−− | Cluster 6 RAR− PA − Sleep + | Cluster 7 RAR − PA− Sleep −− | Cluster 8 RAR − PA + Restless sleep | Cluster 9 RAR −−− PA −− Chronotype − |
|---|---|---|---|---|---|---|---|---|---|---|
| N | 51,507 | 5431 (10.5) | 6494 (12.6) | 8811 (17.1) | 6753 (13.1) | 5522 (10.7) | 8292 (16.1) | 4955 (9.6) | 1746 (3.4) | 3503 (6.8) |
| **Socio-demographic factors** | | | | | | | | | | |
| Age (years), M (SD) | 67.5 (4.2) | 66.3 (3.9) | 67.0 (4.0) | 67.5 (4.1) | 67.0 (4.1) | 67.3 (4.1) | 68.4 (4.3) | 68.1 (4.2) | 67.2 (4.2) | 68.7 (4.3) |
| Sex, women | 27,602 (53.6) | 3330 (61.3) | 3072 (47.3) | 6097 (69.2) | 3294 (48.8) | 3229 (58.5) | 4587 (55.3) | 1632 (32.9) | 951 (54.5) | 1410 (40.3) |
| Non-white ethnicity | 834 (1.6) | 69 (1.3) | 137 (2.1) | 106 (1.2) | 76 (1.1) | 122 (2.2) | 87 (1.0) | 119 (2.4) | 52 (3.0) | 66 (1.9) |
| Lower secondary school or less | 5702 (11.1) | 582 (10.7) | 713 (11.0) | 920 (10.4) | 733 (10.9) | 571 (10.3) | 968 (11.7) | 579 (11.7) | 166 (9.5) | 470 (13.4) |
| Not married/cohabiting | 9139 (17.7) | 839 (15.4) | 1080 (16.6) | 1329 (15.1) | 1164 (17.2) | 1174 (21.3) | 1411 (17.0) | 929 (18.7) | 332 (19.0) | 881 (25.1) |
| In employment | 23,701 (46.0) | 2788 (51.3) | 3309 (51.0) | 3949 (44.8) | 3174 (47.0) | 2617 (47.4) | 3358 (40.5) | 2211 (44.6) | 927 (53.1) | 1368 (39.1) |
| Townsend deprivation index, M (SD) | −2.0 (2.7) | −2.2 (2.5) | −2.1 (2.6) | −2.2 (2.4) | −2.1 (2.6) | −1.7 (2.9) | −2.0 (2.6) | −1.8 (2.8) | −1.9 (2.8) | −1.3 (3.1) |
| **Behavioural factors** | | | | | | | | | | |
| **Smoking** | | | | | | | | | | |
| Never smokers | 27,730 (53.8) | 3117 (57.4) | 3355 (51.7) | 5092 (57.8) | 3912 (57.9) | 2737 (49.6) | 4603 (55.5) | 2359 (47.6) | 939 (53.8) | 1616 (46.1) |
| Ex-smokers | 20,890 (40.6) | 2126 (39.1) | 2775 (42.7) | 3349 (38.0) | 2622 (38.8) | 2349 (42.5) | 3282 (39.6) | 2204 (44.5) | 698 (40.0) | 1485 (42.4) |
| Smokers | 2887 (5.6) | 188 (3.5) | 364 (5.6) | 370 (4.2) | 219 (3.2) | 436 (7.9) | 407 (4.9) | 392 (7.9) | 109 (6.2) | 402 (11.5) |
| **Alcohol consumption** | | | | | | | | | | |
| No consumption | 11,675 (22.7) | 1138 (21.0) | 1258 (19.4) | 1944 (22.1) | 1327 (19.7) | 1329 (24.1) | 2022 (24.4) | 1101 (22.2) | 393 (22.5) | 1163 (33.2) |
| 1–14 units/week | 20,188 (39.2) | 2094 (38.6) | 2466 (38.0) | 3587 (40.7) | 2769 (41.0) | 2151 (39.0) | 3377 (40.7) | 1832 (37.0) | 688 (39.4) | 1224 (34.9) |
| >14 units/week | 19,644 (38.1) | 2199 (40.5) | 2770 (42.7) | 3280 (37.2) | 2657 (39.3) | 2042 (37.0) | 2893 (34.9) | 2022 (40.8) | 665 (38.1) | 1116 (31.9) |
| Fruits and vegetables <twice/day | 4746 (9.2) | 333 (6.1) | 594 (9.1) | 595 (6.8) | 541 (8.0) | 605 (11.0) | 778 (9.4) | 628 (12.7) | 143 (8.2) | 529 (15.1) |
| Daily % time spent outside, M (SD) | 17.2 (13.2) | 20.2 (14.4) | 19.0 (14.1) | 17.6 (13.1) | 17.2 (12.8) | 15.1 (12.2) | 16.0 (12.4) | 17.4 (13.5) | 16.4 (12.9) | 14.8 (12.5) |
| **Health-related factors** | | | | | | | | | | |
| BMI (kg/m²), M (SD) | 26.8 (4.4) | 24.8 (3.5) | 26.2 (3.7) | 26.1 (4.0) | 25.9 (3.6) | 27.5 (4.4) | 27.7 (4.4) | 28.1 (4.6) | 26.8 (4.1) | 29.8 (5.6) |
| Diabetes | 2812 (5.5) | 108 (2.0) | 233 (3.6) | 294 (3.3) | 179 (2.7) | 335 (6.1) | 555 (6.7) | 446 (9.0) | 93 (5.3) | 569 (16.2) |
| Hypertension | 31,638 (61.4) | 2793 (51.4) | 3842 (59.2) | 4984 (56.6) | 3970 (58.8) | 3453 (62.5) | 5499 (66.3) | 3473 (70.1) | 1000 (57.3) | 2624 (74.9) |
| Hyperlipidaemia | 19,691 (38.2) | 1920 (35.4) | 2357 (36.3) | 3584 (40.7) | 2461 (36.4) | 2280 (41.3) | 3408 (41.1) | 1708 (34.5) | 626 (35.9) | 1347 (38.5) |
| Prevalence of ≥1 chronic diseases[a] | 20,344 (39.5) | 1724 (31.7) | 2330 (35.9) | 3299 (37.4) | 2257 (33.4) | 2430 (44.0) | 3551 (42.8) | 2159 (43.6) | 686 (39.3) | 1908 (54.5) |
| CNS medication | 3276 (6.4) | 235 (4.3) | 322 (5.0) | 517 (5.9) | 328 (4.9) | 510 (9.2) | 588 (7.1) | 303 (6.1) | 115 (6.6) | 358 (10.2) |

ADL activity of daily living, BMI body mass index, CNS central nervous system, LIPA light intensity physical activity, M mean, MVPA moderate to vigorous physical activity, PA physical activity, RAR rest-activity rhythm, SD standard deviation.
[a]From a list composed of coronary heart disease, stroke, Parkinson's disease, chronic obstructive pulmonary disease, heart failure, depression, other mental disorders, cancer, liver disease, arthritis.
Value are N (%), unless otherwise stated. Results rounded to one decimal place.

**Table 3 | Association (Odds Ratio (95% CI)) of socio-demographic, seasonal, behavioural, and health-related factors with circadian rhythm clusters in the Whitehall II accelerometer sub-study[a]**

| | Cluster 1 RAR ++ PA ++ | Cluster 2 RAR + LIPA + Sleep − | Cluster 3 RAR + LIPA + Sleep + | Cluster 4 MVPA ++ | Cluster 5 RAR − Chronotype −− | Cluster 6 RAR − PA − Sleep + | Cluster 7 RAR − PA − Sleep −− | Cluster 8 RAR − PA + Restless sleep | Cluster 9 RAR −− PA −− |
|---|---|---|---|---|---|---|---|---|---|
| N (%) | 495 (12.5) | 585 (14.7) | 619 (15.6) | 378 (9.5) | 332 (8.4) | 702 (17.7) | 363 (9.1) | 234 (5.9) | 260 (6.6) |
| **Socio-demographic factors** | | | | | | | | | |
| Age, per 5 years higher age | 0.65 (0.57–0.75)** | 1.04 (0.93–1.17) | Ref | 0.67 (0.58–0.77)** | 0.96 (0.83–1.10) | 1.44 (1.29–1.61)** | 1.48 (1.29–1.69)** | 1.03 (0.89–1.20) | 1.88 (1.61–2.20)** |
| Women | 0.74 (0.55–1.00) | 0.68 (0.51–0.90)** | Ref | 0.31 (0.22–0.45)** | 0.51 (0.35–0.72)** | 0.70 (0.52–0.93)* | 0.29 (0.20–0.43)** | 0.39 (0.26–0.59)** | 0.48 (0.32–0.74)** |
| Non-white ethnicity | 1.44 (0.75–2.76) | 1.92 (1.12–3.29)* | Ref | 0.72 (0.30–1.73) | 2.52 (1.40–4.51)** | 1.21 (0.71–2.07) | 3.78 (2.19–6.53)** | 4.02 (2.18–7.43)** | 2.76 (1.50–5.06)** |
| Lower secondary school or less | 0.95 (0.73–1.23) | 1.25 (0.98–1.59) | Ref | 1.19 (0.90–1.57) | 0.77 (0.57–1.04) | 0.78 (0.61–0.99)* | 0.88 (0.61–1.27) | 1.07 (0.77–1.47) | 1.00 (0.71–1.41) |
| Not married/cohabiting | 1.39 (1.01–1.92)* | 1.03 (0.76–1.39) | Ref | 1.89 (1.35–2.63)** | 2.13 (1.53–2.96)** | 1.04 (0.77–1.39) | 1.48 (1.05–2.07)* | 1.62 (1.10–2.39)* | 1.80 (1.23–2.63)** |
| In employment | 0.76 (0.55–1.05) | 0.89 (0.66–1.21) | Ref | 1.09 (0.79–1.51) | 0.65 (0.45–0.94)* | 0.92 (0.68–1.24) | 0.88 (0.61–1.27) | 1.59 (1.10–2.31)* | 0.41 (0.23–0.73)** |
| **Area index of multiple deprivation** | | | | | | | | | |
| Tertile 1 | Ref | Ref | Ref | Ref | Ref | Ref | Ref | Ref | Ref |
| Tertile 2 | 1.00 (0.74–1.34) | 1.14 (0.87–1.51) | Ref | 0.90 (0.66–1.23) | 0.79 (0.55–1.13) | 0.95 (0.73–1.25) | 0.82 (0.58–1.16) | 1.10 (0.75–1.59) | 1.08 (0.72–1.61) |
| Tertile 3 | 1.35 (0.98–1.85) | 1.59 (1.18–2.13)** | Ref | 1.31 (0.94–1.83) | 1.74 (1.23–2.46)** | 1.02 (0.76–1.37) | 1.61 (1.14–2.28)** | 1.50 (1.01–2.22)* | 1.28 (0.84–1.95) |
| **Behavioural factors** | | | | | | | | | |
| **Smoking** | | | | | | | | | |
| Never smokers | Ref | Ref | Ref | Ref | Ref | Ref | Ref | Ref | Ref |
| Ex-smokers | 1.10 (0.86–1.43) | 0.98 (0.78–1.25) | Ref | 1.16 (0.89–1.52) | 1.30 (0.97–1.74) | 0.91 (0.72–1.15) | 0.93 (0.70–1.23) | 1.09 (0.79–1.50) | 1.00 (0.71–1.40) |
| Current smokers | 0.54 (0.19–1.49) | 2.44 (1.11–5.34)* | Ref | 0.91 (0.32–2.61) | 5.44 (2.45–12.10)** | 2.67 (1.19–5.95)* | 3.10 (1.28–7.50)* | 0.95 (0.29–3.19) | 8.76 (3.59–21.35)** |
| **Alcohol consumption** | | | | | | | | | |
| No consumption | Ref | Ref | Ref | Ref | Ref | Ref | Ref | Ref | Ref |
| 1–14 units/week | 0.95 (0.66–1.35) | 1.01 (0.73–1.39) | Ref | 1.34 (0.90–2.00) | 0.80 (0.56–1.16) | 0.78 (0.57–1.05) | 0.89 (0.62–1.29) | 0.86 (0.56–1.31) | 0.76 (0.51–1.14) |
| > 14 units/week | 1.37 (0.92–2.05) | 1.35 (0.93–1.95) | Ref | 1.55 (0.99–2.41) | 1.00 (0.65–1.53) | 0.83 (0.58–1.20) | 1.09 (0.71–1.67) | 1.35 (0.84–2.17) | 0.58 (0.34–0.97)* |
| Fruits and vegetables, <twice a day | 1.19 (0.85–1.66) | 1.15 (0.85–1.56) | Ref | 0.77 (0.53–1.11) | 1.24 (0.88–1.77) | 1.22 (0.90–1.65) | 1.67 (1.20–2.34)** | 1.30 (0.89–1.91) | 1.62 (1.09–2.39)* |
| Daily % >1000 lux, 1 SD higher exposure | 1.82 (1.61–2.05)** | 1.05 (0.93–1.18) | Ref | 0.92 (0.79–1.06) | 0.52 (0.44–0.63)** | 0.51 (0.44–0.58)** | 0.48 (0.40–0.57)** | 1.17 (0.998–1.36) | 0.21 (0.16–0.28)** |
| **Health-related factors** | | | | | | | | | |
| BMI, per 5 kg/m² higher BMI | 0.70 (0.59–0.83)** | 1.39 (1.19–1.62)** | Ref | 0.90 (0.75–1.09) | 1.55 (1.30–1.85)** | 1.73 (1.49–2.00)** | 2.31 (1.95–2.75)** | 1.13 (0.92–1.40) | 2.78 (2.30–3.37)** |
| Diabetes | 0.70 (0.43–1.14) | 1.05 (0.72–1.54) | Ref | 0.52 (0.30–0.90)* | 0.90 (0.57–1.42) | 1.37 (0.96–1.96) | 1.23 (0.82–1.85) | 0.71 (0.41–1.23) | 1.32 (0.84–2.06) |
| Hypertension | 0.87 (0.67–1.13) | 1.30 (1.01–1.66)* | Ref | 0.95 (0.72–1.25) | 0.77 (0.57–1.03) | 1.27 (0.99–1.62) | 0.95 (0.70–1.29) | 1.07 (0.77–1.49) | 1.26 (0.86–1.86) |
| Hyperlipidaemia | 0.87 (0.67–1.13) | 1.03 (0.81–1.31) | Ref | 0.93 (0.71–1.22) | 0.87 (0.65–1.16) | 1.00 (0.79–1.26) | 1.03 (0.77–1.37) | 1.08 (0.79–1.48) | 1.06 (0.76–1.48) |
| Prevalence of ≥1 chronic diseases[b] | 0.94 (0.72–1.24) | 0.98 (0.76–1.26) | Ref | 0.94 (0.70–1.26) | 1.37 (1.01–1.86)* | 1.04 (0.81–1.33) | 1.11 (0.82–1.49) | 0.87 (0.62–1.23) | 1.53 (1.08–2.18)* |
| CNS medication | 0.44 (0.23–0.86)* | 1.10 (0.65–1.85) | Ref | 0.56 (0.27–1.19) | 1.71 (0.98–2.98) | 1.21 (0.73–2.02) | 2.52 (1.45–4.37)** | 2.38 (1.31–4.32)** | 3.57 (2.01–6.33)** |

ADL activity of daily living, BMI body mass index, CI confidence interval, CNS central nervous system, LIPA light intensity physical activity, lux illuminance, M mean, MVPA moderate to vigorous physical activity, PA physical activity, RAR rest-activity rhythm, SD standard deviation.

*P < 0.05, **P < 0.01.

[a]Multinomial logistic regression was performed to compute the odds ratios, the corresponding 95% CIs, and p-values (source data file). All p-values were two-sided with a significance level set at 0.05. Cluster 3 was used as the reference, and the model was mutually adjusted for all presented variables as well as the season of wear.

[b]From a list composed of coronary heart disease, stroke, Parkinson's disease, chronic obstructive pulmonary disease, heart failure, depression, other mental disorders, cancer, liver disease, and arthritis.

**Table 4 | Association (Odds Ratio (95% CI)) of socio-demographic, seasonal, behavioural, and health-related factors with circadian rhythm clusters in the UK Biobank accelerometer sub-study[a]**

| | Cluster 1 RAR ++ PA ++ | Cluster 2 RAR + PA + Sleep - | Cluster 3 RAR + LIPA + Sleep + | Cluster 4 MVPA ++ | Cluster 5 RAR - Chronotype -- | Cluster 6 RAR - PA - Sleep + | Cluster 7 RAR - PA - Sleep -- | Cluster 8 RAR + PA + Restless sleep | Cluster 9 RAR -- PA -- Chronotype - |
|---|---|---|---|---|---|---|---|---|---|
| N | 5431 (10.5) | 6494 (12.6) | 8811 (17.1) | 6753 (13.1) | 5522 (10.7) | 8292 (16.1) | 4955 (9.6) | 1746 (3.4) | 3503 (6.8) |
| **Socio-demographic factors** | | | | | | | | | |
| Age, per 5 years higher age | 0.63 (0.60–0.66)** | 0.83 (0.79–0.87)** | Ref | 0.79 (0.76–0.83)** | 0.96 (0.92–1.01) | 1.27 (1.22–1.33)** | 1.17 (1.12–1.23)** | 0.98 (0.91–1.05) | 1.44 (1.36–1.52)** |
| Women | 0.63 (0.58–0.68)** | 0.40 (0.37–0.43)** | Ref | 0.38 (0.36–0.41)** | 0.61 (0.57–0.66)** | 0.55 (0.52–0.59)** | 0.23 (0.21–0.25)** | 0.54 (0.48–0.60)** | 0.28 (0.26–0.31)** |
| Non-white ethnicity | 1.00 (0.73–1.36) | 1.74 (1.34–2.25)** | Ref | 0.89 (0.66–1.20) | 1.67 (1.28–2.19)** | 0.83 (0.62–1.10) | 1.93 (1.47–2.54)** | 2.24 (1.59–3.14)** | 1.27 (0.92–1.76) |
| Lower secondary school or less | 1.24 (1.11–1.39)** | 1.10 (0.99–1.23) | Ref | 1.18 (1.06–1.31)** | 0.90 (0.80–1.01) | 0.98 (0.88–1.08) | 0.96 (0.86–1.02) | 0.91 (0.76–1.09) | 0.91 (0.80–1.04) |
| Not married/cohabiting | 1.12 (1.02–1.24)* | 1.28 (1.17–1.40)** | Ref | 1.35 (1.23–1.47)** | 1.49 (1.37–1.64)** | 1.18 (1.09–1.29)** | 1.50 (1.36–1.65)** | 1.38 (1.20–1.58)** | 1.85 (1.66–2.05)** |
| In employment | 0.98 (0.91–1.06) | 1.09 (1.01–1.17)* | Ref | 0.88 (0.82–0.95)** | 0.98 (0.91–1.06) | 0.93 (0.86–0.99)* | 0.98 (0.90–1.07) | 1.26 (1.12–1.42)** | 0.87 (0.79–0.96)** |
| **Townsend deprivation index** | | | | | | | | | |
| Tertile 1 | Ref | Ref | Ref | Ref | Ref | Ref | Ref | Ref | Ref |
| Tertile 2 | 0.97 (0.89–1.05) | 1.00 (0.93–1.08) | Ref | 0.96 (0.89–1.04) | 0.96 (0.89–1.05) | 0.99 (0.92–1.07) | 1.03 (0.94–1.13) | 0.97 (0.86–1.11) | 0.99 (0.89–1.10) |
| Tertile 3 | 0.94 (0.86–1.03) | 1.09 (1.01–1.19)* | Ref | 1.10 (1.02–1.19)* | 1.33 (1.22–1.45)** | 1.17 (1.08–1.26)** | 1.36 (1.24–1.48)** | 1.22 (1.07–1.39)** | 1.59 (1.44–1.77)** |
| **Behavioural factors** | | | | | | | | | |
| **Smoking** | | | | | | | | | |
| Never smokers | Ref | Ref | Ref | Ref | Ref | Ref | Ref | Ref | Ref |
| Ex-smokers | 1.09 (1.01–1.17)* | 1.14 (1.07–1.23)** | Ref | 0.96 (0.90–1.03) | 1.18 (1.10–1.27)** | 0.94 (0.88–1.01) | 1.07 (0.99–1.16) | 1.04 (0.93–1.16) | 1.01 (0.93–1.11) |
| Smokers | 0.71 (0.59–0.86)** | 1.19 (1.02–1.38)* | Ref | 0.64 (0.54–0.76)** | 1.87 (1.61–2.17)** | 1.14 (0.98–1.32) | 1.71 (1.47–2.00)** | 1.36 (1.08–1.71)** | 2.65 (2.25–3.12)** |
| **Alcohol consumption** | | | | | | | | | |
| No consumption | Ref | Ref | Ref | Ref | Ref | Ref | Ref | Ref | Ref |
| 1–14 units/week | 0.95 (0.86–1.04) | 1.03 (0.94–1.13) | Ref | 1.07 (0.98–1.17) | 0.94 (0.86–1.03) | 0.95 (0.87–1.03) | 0.93 (0.84–1.02) | 0.97 (0.84–1.11) | 0.68 (0.61–0.75)** |
| > 14 units/week | 1.07 (0.98–1.18) | 1.24 (1.14–1.36)** | Ref | 1.13 (1.03–1.23)** | 0.97 (0.88–1.06) | 0.90 (0.83–0.97)* | 1.10 (0.99–1.21) | 1.03 (0.89–1.18) | 0.68 (0.61–0.76)** |
| Fruits and vegetables <twice/day | 0.83 (0.72–0.96)** | 1.11 (0.99–1.26) | Ref | 0.99 (0.88–1.12) | 1.41 (1.25–1.60)** | 1.24 (1.11–1.39)** | 1.42 (1.25–1.60)** | 1.04 (0.86–1.26) | 1.68 (1.47–1.92)** |
| Daily % >1000 lux, per 1 SD higher exposure | 1.27 (1.22–1.32)** | 1.08 (1.04–1.12)** | Ref | 0.99 (0.96–1.03) | 0.81 (0.78–0.85)** | 0.82 (0.79–0.85)** | 0.87 (0.83–0.91)** | 0.94 (0.88–0.996)* | 0.68 (0.64–0.71)** |
| **Health-related factors** | | | | | | | | | |
| BMI, per 5 km/m² higher BMI | 0.58 (0.55–0.61)** | 0.95 (0.91–0.99)* | Ref | 0.87 (0.83–0.91)** | 1.41 (1.35–1.47)** | 1.50 (1.45–1.56)** | 1.62 (1.55–1.69)** | 1.20 (1.12–1.28)** | 2.13 (2.03–2.23)** |
| Diabetes | 0.77 (0.62–0.97)* | 0.93 (0.77–1.11) | Ref | 0.74 (0.61–0.90)** | 1.26 (1.06–1.48)** | 1.34 (1.15–1.56)** | 1.50 (1.28–1.75)** | 1.30 (1.01–1.66)* | 2.19 (1.87–2.57)** |
| Hypertension | 1.01 (0.93–1.08) | 1.05 (0.98–1.13) | Ref | 1.09 (1.02–1.17)* | 1.09 (1.02–1.18)* | 1.14 (1.07–1.22)** | 1.19 (1.10–1.29)** | 0.93 (0.84–1.04) | 1.27 (1.15–1.39)** |
| Hyperlipidaemia | 0.96 (0.89–1.03) | 1.03 (0.96–1.10) | Ref | 1.03 (0.96–1.10) | 1.08 (1.00–1.16)* | 1.08 (1.01–1.15)* | 1.04 (0.96–1.12) | 0.92 (0.82–1.03) | 1.12 (1.03–1.23)** |
| Prevalence of ≥1 chronic diseases[b] | 0.90 (0.84–0.97)** | 0.98 (0.92–1.06) | Ref | 0.90 (0.84–0.96)** | 1.17 (1.08–1.26)** | 1.09 (1.02–1.16)* | 1.13 (1.05–1.22)** | 1.07 (0.95–1.19) | 1.43 (1.31–1.56)** |
| CNS medication | 0.85 (0.72–1.00) | 0.97 (0.84–1.13) | Ref | 1.02 (0.87–1.18) | 1.42 (1.24–1.63)** | 1.18 (1.03–1.34)* | 1.11 (0.94–1.29) | 1.18 (0.94–1.47) | 1.37 (1.17–1.61)** |

*BMI* body mass index, *CI* confidence interval, *CNS* central nervous system, *lux* illuminance, *M* mean, *LIPA* light intensity physical activity, *PA* physically active, *MVPA* moderate to vigorous physical activity, *RAR* rest-activity rhythm, *SD* standard deviation.
*P < 0.05, **P < 0.01.
[a]Multinomial logistic regression was performed to compute the odds ratios, the corresponding 95% CIs, and p-values (source data file). All p-values were two-sided with a significance level set at 0.05. Cluster 3 was used as the reference, and the model was mutually adjusted for all presented variables as well as season of wear.
[b]From a list composed of coronary heart disease, stroke, Parkinson's disease, chronic obstructive pulmonary disease, heart failure, depression, other mental disorders, cancer, liver disease, and arthritis.

Our two-step approach (PCA for data reduction followed by cluster analyses) captures the behavioural manifestation of the endogenous biological clock by capturing multiple aspects of the sleep-wake cycle using data from two population-based cohort studies. Accelerometers provide scalable, cost-effective measures of the sleep-wake cycle—one of the most visible manifestations of the circadian clock. It remains important to consider the influence of extrinsic factors, such as seasonal variations or activity routines, that may also influence individual behaviours. For example, we found employment status to affect the clusters identified in our study, as also demonstrated in another study where work environment and schedule impacted sleep[30]. Exposure to zeitgebers such as light plays an important role in the synchronisation of the internal clock, and misalignment of zeitgebers with endogenous biological clock can lead to chronic perturbations of circadian rhythm[31]. Further studies are needed to determine the association between behavioural manifestation of the biological clock and its molecular and genetic markers.

In our study the size of the nine circadian rhythm clusters was slightly different in WII and UKB, possibly due to difference in the distribution of age, sex, and education as well as differences in accelerometer protocol. However, in both studies participants' sociodemographic, behavioural, and health-related factors differed across the clusters. Previous studies have reported participants' characteristics to vary across metrics such as interdaily stability and intradaily variability[16,32,33], relative amplitude, and $M_{10}$ (most active 10 h period) and $L_5$ (least active 5 h period)[16]. Older age has been shown to be associated with a more stable[32,33] but fragmented RAR[5,16,32]—as denoted by higher interdaily stability and intradaily variability—and less daytime activity[16]. Women and people married/cohabiting have been shown to have more stable and less fragmented rhythm[32,33]. Smoking has also been found to be associated with a less stable rhythm[16,32,33] while higher BMI was associated with less stable but also more fragmented rhythm[16,32].

Our results add to the existing evidence by showing heterogeneity in associations with sociodemographic, behavioural and health-related factors for a given level of RAR depending on the other dimensions of circadian rhythm—daytime activity, sleep, and chronotype. We found women and persons married/cohabiting to have a more stable and less fragmented rhythm, undertake light intensity physical activity, and have good sleep levels. Living in a more deprived area was more common in clusters characterised by sleep disturbances and delayed chronotype in WII, whereas in UKB it was more common in clusters with poorer RAR. Participants with greater outdoor light exposure had more robust RAR and were likely to be active, independently of the season, as previously shown in UKB using self-reported time spent outdoors[34]. As the number of blind participants was small (< 0.01% in WII and UKB), this group is unlikely to influence findings on light exposure. Participants in clusters characterised by poor sleep and/or disturbed RAR were more likely to be on CNS medications. Poor health status indicated by prevalent diabetes or chronic diseases tended to be more frequent in participants with poorer RAR and less physical activity.

Strengths of our study include objective measurement of circadian rhythm, the large number of metrics covering four behavioural circadian rhythm dimensions, statistical analysis in two cohort studies, use of open-source software to derive metrics, and analyses on a large set of covariates to study their association with circadian rhythm clusters. This study also has limitations. First, both cohort studies included mainly white participants in the UK and whether circadian rhythm differs in other countries remains unknown. Second, a sleep diary was not used in UKB, potentially leading to some misclassification of sleep and waking time due to inaccuracies in estimation of sedentary behaviour, sleep, and chronotype metrics, particularly if participants were sedentary before sleep[35]. Three, in UKB some covariates were measured at inclusion rather than concurrently with the accelerometer measure but a previous study in UKB did not find major differences in associations for covariates drawn from baseline or at the same time as the accelerometer[36]. In addition, the pattern of results in our study was similar in WII and UKB, suggesting little impact of when covariates were measured. Fourth, in the absence of a consensus on measuring naps, we did not use data on naps in the identification of circadian rhythm clusters. We recognise that naps are likely to contribute to intradaily variability, transition probabilities from activity to rest, and sedentary behaviour.

Using data from two large, population-based cohort studies, our study demonstrates the clustering in behavioural circadian rhythm metrics, resulting in participants to be grouped in nine distinct clusters. Despite the RAR metrics being similar across several clusters, our approach shows considerable differences in the distribution of daytime activity, sleep, and chronotype across the clusters. Not considering these differences in studies that focus on RAR does not allow accurate measurement of circadian rhythm. Our analyses show a number of sociodemographic, behavioural and health-related factors to differ across these clusters. Given the dysregulation of circadian rhythm at older ages and in certain chronic diseases, the findings of the present study highlight the need for better characterisation of circadian rhythm.

## Methods

### Study populations

The WII study is an ongoing prospective cohort study established in 1985–1988 on 10,308 British civil servants, consisting of a clinical examination every 4 to 5 years. An accelerometer measure was added to the 2012–2013 wave of data collection for 4880 men and women (aged 60–83 years) seen at the London clinic and those living in the South-Eastern regions of England who underwent clinical examination at home[23]. Written informed consent for participation was obtained at each contact. Research ethics approval was obtained from the University College London ethics committee at each wave (latest reference number 85/0938).

The UKB is a prospective population-based cohort study on over 500,000 men and women aged between 40 and 69 years, registered with the UK National Health Service. Baseline measurements took place between 2006 and 2010. The accelerometer sub-study was undertaken on 236,488 individuals between 2013–2015[37]. Participants aged ≥60 at accelerometer sub-study were retained in the present analysis as our focus was circadian rhythm clusters at older ages. Approval was received from the National Information Governance Board for Health and Social Care and the National Health Service North West Centre for Research Ethics Committee (reference number 11/NW/0382). All participants gave written consent for participation. Access to UKB data in our study was under application number 96856.

### Accelerometer wear protocols

In WII, individuals wore a GENEActiv (Activinsights Ltd, Kimbolton, UK) triaxial accelerometer on their non-dominant wrist and completed a sleep diary over 24 h, for nine consecutive days. Accelerometer data were sampled at 85.7 Hz with the acceleration expressed relative to gravity ($1g = 9.81$ m/s$^2$), and processed using GGIR R package version 3.1–7[38]. Raw acceleration was calculated with the metric Euclidean Norm Minus One (ENMO), negative values rounded to zero. Then these values were corrected for calibration error and non-wear time[39]. Data from days 2 to 9 (penultimate day) were retained in the analysis, comprising a total of seven consecutive full day windows (daytime waking period and following sleep period). Sleep onset and waking up times were detected using an algorithm, guided by a sleep diary[39]. The daytime waking periods were defined as the time between waking up to start the day and sleep onset at night, and sleep periods between sleep onset and waking up to start the day. Participants were included in the analyses if they had data from at least five valid full day windows

(defined as wear times ≥2/3 of both waking and following sleep periods)[40].

In UKB, participants wore an Axivity AX3 triaxial accelerometer on their dominant wrist over 24 h for seven consecutive days, starting at 10 h on day 1[37]. Data were sampled at 100 Hz. As a sleep diary was not used, sleep periods were detected using an algorithm that has been described previously[39]. To maximise measures covering full day windows (sleep period and following waking period), data from the sleep onset of the first night until sleep onset on the last night were retained in the analyses, comprising a total of six consecutive full day windows. Participants were included in our analyses if they had data on at least five valid full day windows[40].

### Measures of behavioural circadian rhythm

A total of 36 accelerometer-assessed metrics (Supplementary Table S1) were derived to measure four dimensions of circadian rhythm—RAR, daytime activity, sleep, and chronotype.

RAR (6 metrics) was based on the day windows and included parametric measures based on the cosinor function fitted to the log of the acceleration signal[41], that yielded three metrics: mesor (average activity according to the function), amplitude (the peak of the function minus the mesor), and $R^2$ (goodness of fit)[41]. The three non-parametric metrics were: interdaily stability, intradaily variability[5], and the relative amplitude based on the mean acceleration of $M_{10}$ and the least active 5 h period ($L_5$)[42].

Daytime activity (15 metrics) was based on activity levels during waking periods and classified as SB, LIPA, and MVPA[43]. For each activity level, we calculated the daytime total duration, number and mean duration of bouts. We also considered: mean acceleration during waking period and during $M_{10}$, transition probabilities (TPs) to switch from activity levels (LIPA or MVPA) to rest (SB), and from rest to activity, during the waking period ($TP_{ar,d}$ and $TP_{ra,d}$)[5], and two parameters derived from the acceleration distribution (intensity gradient (IG) intercept, and slope)[44].

Sleep (10 metrics) included: total sleep duration, sleep efficiency during the sleep period, mean acceleration during sleep and during $L_5$, number and mean duration of sleep bouts, TPs from sleep to wake, or wake to sleep during the night ($TP_{sw,n}$ and $TP_{ws,n}$), duration of wake after sleep onset (WASO), and mean duration of wake bouts.

Chronotype (5 metrics) was characterised by: time of sleep onset and waking, starting time of $M_{10}$ and $L_5$, and the cosinor acrotime (timing of the peak of the function).

All metrics were derived using GGIR R package version 3.1–7.

### Participants' characteristics

In WII, covariates were measured at the 2012–2013 wave either using questionnaire, clinical examination, or data from electronic health records, including the Hospital Episode Statistics (HES) and Mental Health Services Data Set. In UKB, covariates were either extracted from the baseline examination in 2006–2010 or extracted from electronic health records (HES).

Socio-demographic factors included age at accelerometer data collection, sex (defined based on the British civil servants record in WII, and from the UK National Health Service, updated in some cases by self-report of the participant, in UK Biobank), ethnicity (white or non-white), education (lower secondary school or less, secondary school, or higher than secondary school), cohabitation status (married/cohabiting or not married/cohabiting), whether in employment (yes or no), and area deprivation index (Index of multiple deprivation in WII[45] and Townsend deprivation index in UKB[46]).

Behavioural factors included smoking status (never, ex-, or current-smoker), alcohol consumption (0, 1–14, or >14 units/week), fruit and vegetable consumption (less than or ≥ twice daily), and exposure to daylight, estimated as the proportion of waking time exposed to >1000 lux based on the light sensor from the GENEActiv accelerometer

in WII[47] and self-report of the time spent outdoors in UKB[34]. The light measures were standardised to account for season of accelerometer measurement. In additional analyses, we used self-reported chronotype preference in UKB, categorised as: Definitely a morning person, More a morning than evening person, More an evening than a morning person, and Definitely an evening person. Data on shift work were only available in UKB and were used in additional analyses, categorised as: No and Yes.

Health-related factors included BMI (in kg/m², calculated from measures of height and weight at the clinical examination), diabetes (measured using fasting glucose at the clinical examination ≥7.0 mmol/L, self-reported doctor-diagnosed diabetes, use of anti-diabetic medications, or record in HES), hypertension (defined as systolic/diastolic blood pressure ≥140/90 mmHg, use of anti-hypertensive drugs, or record in HES), hyperlipidaemia (defined as low-density lipoproteins >4.1 mmol/L or use of lipid-lowering drugs), chronic disease (yes, no out of coronary heart disease, stroke, Parkinson's disease, chronic obstructive pulmonary disease, heart failure, depression, other mental disorders, cancer, liver disease, and arthritis, and extracted from multiple sources including HES records, mental health records, cancer registry, and data collected at the Whitehall clinical examinations in WII, and from HES, clinical measures, and medication data in UKB), use of CNS medication (among anti-depressant, antipsychotic, hypnotic, anxiolytic, or Parkinson medications), reported by participant in the section on medication in the questionnaire.

Mortality data were used in preliminary analysis. Mortality cases were ascertained using linkage to the National Health Service (NHS) mortality register in both cohort studies. Participants were followed from date of accelerometer wear to November 2023 in WII and November 2022 in UKB.

### Statistical analysis

To justify the use of 36 rather than the 10 commonly used metrics (interdaily stability, intradaily variability, relative amplitude, cosinor mesor, cosinor amplitude, cosinor acrotime, $M_{10}$ and $L_5$ timing and mean acceleration[6,8,48–50]) to characterise circadian rhythm we compared the predictive performance of these two sets of metrics for risk of mortality. We first used principal component analysis (PCA) to extract components to reflect the metrics too strongly correlated to be used in the prediction analysis. The number of components for each set of metrics (10 and 36) was selected using the eigenvalue criterion (≥1) and the cumulated percentage of variance explained (at least 75%). Cox regression analysis was used to examine the association of these components with mortality, and their predictive performances were compared using the C-index.

As the predictive performance of components using 36 metrics was better, subsequent analyses were conducted using the set of 36 metrics in parallel in both cohort studies to identify how circadian rhythm metrics clustered in individuals, and then examine the association of the clusters with covariates. The correlation between the 36 metrics (reflecting RAR, daytime activity, sleep, and chronotype) was examined using Pearson's correlation coefficients. Then, to examine how the 36 metrics were distributed across individuals we undertook cluster analyses; the workflow for steps in these analyses is shown in Supplementary Fig. S3. The procedure to identify clusters in both WII and UKB was made in three steps. First, standardisation of the 36 metrics using *scale* function to make them comparable. Second, we applied the data reduction method PCA on this standardised metrics using *prcomp* function. Using eigenvalue ≥1 and cumulative variance explained ≥75% of the variance of the 36 metrics, we selected N component. Finally, we used k-means clustering with the *kmeans* function on the retained components, testing for K = 4 to 12, with 5000 numbers of random starts, and 10,000 maximum number of iterations. To determine the number of cluster K retained, we examined statistical

criteria—within cluster sum of squares, Silhouette coefficient, and Davies-Bouldin index—to assess the statistical differences between clustering solutions. As the differences were small, the interpretability of the clusters was further examined. For every K (number of clusters) solution, we looked at the visualisation of the solutions to interpret each cluster. The aim was to determine whether the addition of a new cluster provided new relevant information in terms of differences in the circadian rhythm profiles. We also considered cluster sizes, ensuring that each contained at least $N \geq 200$ participants to be suitable for analysis.

In WII, the assumptions for post-hoc Tukey tests were not met and to allow interpretation of the clusters as a function of each metric we used the non-parametric simultaneous rank test procedure. In UKB, the large sample size did not allow non-parametric simultaneous rank test to be used, leading us to use the post-hoc Tukey test with ANOVA[51].

After identification of circadian rhythm clusters, we used multinomial logistic regression to examine associations of participants' characteristics and season of wear (independent variables) with circadian rhythm clusters (dependent variable). The reference category in the multinomial regression was chosen based on size (number of participants), the cluster where none of the metric means was above or below one SD from the population mean, and better health status.

### Reporting summary
Further information on research design is available in the Nature Portfolio Reporting Summary linked to this article.

## Data availability
Data cannot be made publicly available because of the risk of participant re-identification, and the study coordinators' engagement to only share their data for research purposes due to ethics and IRB restrictions. However, a data sharing portal allows access to data to undertake analyses within the secure portal in WII https://portal.dementiasplatform.uk/. The UK Biobank data are available through a procedure described at https://www.ukbiobank.ac.uk/enable-your-research. Source data are provided with this paper.

## Code availability
All analytical codes are shared on GitHub at https://github.com/samvidil/Clusters-of-circadian-rhythm-article, in order to reproduce all Tables and figures from the article. Additionally, the codes can be found at Zenodo repository through https://doi.org/10.5281/zenodo.17417859[52].

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

## Acknowledgements

We thank all of the participating civil service departments and their welfare, personnel, and establishment officers; the British Occupational Health and Safety Agency; the British Council of Civil Service Unions; all participating civil servants in the Whitehall II study and participants in UK Biobank; and all members of the Whitehall II study team and UK Biobank team. The Whitehall II study is and was supported by grants from the National Institute on Aging, NIH (R01AG056477, R01AG062553); UK Medical Research Council (R024227, S011676), and the Wellcome Trust (221854/Z/20/Z). SS is funded by the European Union (ERC grant number 101043884), the Fondation Alzheimer and the Fondation Vaincre Alzheimer, ASM by France 2030 ANR-23-PAVH-0006. The funding agencies had no role in the study design, data collection, analyses, and interpretation of the data or writing of the manuscript. Views and opinions expressed are, however, those of the authors only and do not necessarily reflect those of the funding agencies. Neither the European Union nor the granting authority can be held responsible for them.

## Author contributions

Conceptualisation: S.V. and S.S. Methodology: S.V., I.M.D., and S.S. Formal analysis: S.V. Access and verified the data: S.V., I.M.D., V.V.H., and S.S. Data Curation: V.V.H., I.M.D., and A.F. Writing—original draft preparation: S.V. and S.S. Writing—review and editing: S.V., I.M.D., A.D., A.F., B.L., M.C., V.V.H., A.S.M., and S.S. Supervision: I.M.D. and S.S. Funding acquisition: A.S.M. and S.S.

## Competing interests

The authors declare no competing interests.
