## [Peer Review File · Nature Communications]

Circadian rhythm profiles derived from accelerometer measures of the sleep-wake cycle in two cohort studies

Corresponding Author: Dr Severine Sabia

Version 0:

Reviewer comments:

Reviewer #1

(Remarks to the Author)

This study uses a clustering approach to identify nine distinct circadian rhythm phenotypes based on accelerometer-based metrics from the Whitehall II study (n=3991) and UK Biobank (n=54,995). These phenotypes are categorized by combinations of rest-activity rhythm (RAR), daytime activity, sleep efficiency and chronotype metrics. The derived phenotypes were found to be associated with a number of demographic, behavioural and health traits including age, smoking, light exposure, BMI, depressive symptoms and medication use. The work appears original in its methodology of combining four dimensions of circadian rhythm and identifying a variety of phenotypes. However, the proposed value and significance of identifying the new multi-faceted phenotypes is unclear. I was not convinced that the work would be of sufficient interest or significance to the field, and proposed links between the phenotypes and health outcomes was not clearly demonstrated. More specific comments and questions are outlined below:

ABSTRACT

- 1) The method/methods used to identify circadian rhythm phenotypes are not described
- 2) What is meant by 'opposite RAR and daytime activity'?
- 3) What is the distinction between inefficient and restless sleep?
- 4) What is value of identifying composite phenotypes rather than the 4 conceptual dimension and how interpretable/meaningful are the associations identified?

INTRODUCTION

- 5) Explain distinction (and correlation) between the 4 different conceptual dimensions
- 6) Can accelerometers identify true chronotype as defined by 'natural preference'?

RESULTS

- 7) The range of correlations between metrics should be included in the text.
- 8) Some terms were not defined in the Results and would aid interpretation e.g. IG, mesor, relative amplitude
- 9) When were the characteristics assessed with respect to accelerometer wear?
- 10) It is difficult to get sense of magnitude/statistical robustness for associations between characteristics and CR phenotypes, hindering interpretation
- 11) Inference from season difficult as presumably the date the participants were given accelerometer devices were largely random, and there were no repeated measures between seasons?
- 12) Mention 36 metrics for each cluster but only RAR, PA, SB, LIPA, Sleep, late chronotype, restless sleep described - why? How do these map to the 4 conceptual dimensions?
- 13) Describe in more detail the cluster based analysis to get from 8 PCs to 9 clusters/phenotypes
- 14) I'm struggling to understand the utility/purpose of classifying these composite phenotypes rather than grouping based on conceptual dimensions. The authors mention that use of conceptual dimensions may not classify as accurately, but it would be useful to demonstrate/quantify this
- 15) The results of replication in UKB quite poorly described
- 16) More description of e.g. proportion of people within the different phenotypic groups in the different populations would be useful
- 17) Were/can the clusters from WHII be directly overlayed into UKB and was goodness of fit assessed?
- 18) Mention is made to differences in some phenotype differences e.g. fruit and veg consumption in Discussion but not in Results

DISCUSSION

- 19) The authors state "As some covariates were not available in the UK Biobank study, we could not undertake a strict

replication of the analysis on associated factors” - is this referring to characteristics and/or sleep metrics? A better attempt at assessing consistency in associations between the studies could be made for those common variables that do exist

METHODS

20) The authors describe how the cluster closest to the population mean is used as the reference category for multinomial regression in methods - how was this determined?

21) What was the exposure and what was outcome in the association analysis with characteristics - this should be more clearly articulated.

22) In multinomial regression, were all variables mutually adjusted and can this be justified?

23) Methods for reporting UKB are too brief - ‘we repeated analyses with some modifications’

24) The decision and justification of restricting UKB individuals to only those over age of 60 needs to be explained

25) It is unclear why Whitehall II is treated as the primary analysis when it is >10x smaller than UKB

(Remarks on code availability)

Authors state that code will be shared on GitHub upon publication of the paper so I have been unable to review this.

Reviewer #2

(Remarks to the Author)

I believe this work has significant merit, but certain aspects would benefit from further clarification. Below, I outline several points that may help refine the interpretation of the findings and improve the overall clarity of the study.

Considering the associations in Tables 2 and 3, the relationships with the nine circadian rhythm phenotypes illustrate how both extrinsic and intrinsic factors shape the timing and intensity of behaviours across 24 hours. To what extent are certain circadian rhythm phenotypes primarily driven by external factors? Do they represent true circadian phenotypes (i.e., behaviours that persist under constant routine conditions), or do they simply reflect discrepancies between an individual's preferred behavioural timing and the constraints imposed by life circumstances (e.g., parenting responsibilities, work schedules, or leisure activity timing)?

If I'm not mistaken, participants—at least in the UK Biobank—were asked about their circadian preference (chronotype). How prevalent are the subjective chronotype subtypes within each of your nine identified phenotypes?

You found that daily light exposure had a strong association with the odds of being assigned to a certain circadian phenotype. Did you exclude individuals with blindness (e.g., see Data-Field 131212)?

Given the association between both age and female sex with the circadian phenotypes, I wonder if there is an interaction between female sex and age.

Why was napping not considered in the construction of your circadian phenotypes?

Minor correction: In Table 2, you stated that the reference group consisted of 625 subjects.

I've checked the prevalence of the various circadian phenotypes in the Whitehall Study and UK Biobank study. They differ somewhat. However, both cohorts are from the UK, and the data were collected at different times, correct? Does this suggest that not only seasons are associated with certain circadian phenotypes, but also that longer biological time scales may have an impact?

Given the potential impact of extrinsic factors (e.g., noise), I wonder if you could expand your socio-demographic analysis to include the area of residence.

(Remarks on code availability)

Reviewer #3

(Remarks to the Author)

The manuscript describes a study to use wrist worn accelerometer data to identify new circadian rhythm phenotypes. Thirty metrics, PCA and clustering are used in the Whitehall II study to derive these phenotypes. Associations with a range of traits and conditions is then performed. Replication of the circadian rhythm dimensions are performed in UK Biobank and associations are performed in UK Biobank.

This is a nicely written study with some interesting data. I do have some concerns though.

My major concern is that I am not clear how the authors have demonstrated that these 9 metrics are measuring circadian phenotypes. PCA and clustering analyses are performed but it not clear why the authors think these are measuring specifically circadian phenotypes. The fact that the clusters appear to be similarly recreated in UK Biobank is encouraging - suggesting it is not just an artefact of clustering. But why the authors are sure that these 9 phenotypes represent new circadian features needs to be made clearer to me.

Correlation analyses against a range of phenotypes is performed and the new circadian phenotypes appear to associate

with what might be expected. It is difficult to make too much of these epidemiological analyses though and I wonder if genetics could be used to identify whether these seem to be reasonable clusters as has been done for other diseases (e.g. <https://link.springer.com/article/10.1007/s00125-022-05848-6>).

I have an additional concern that, at least for the UK Biobank, activity monitors were worn some times several years after the baseline measurements. Does this affect interpretation of any of the association results?

(Remarks on code availability)

Version 1:

Reviewer comments:

Reviewer #1

(Remarks to the Author)

Overall comments

The authors have made a strong attempt to improve the manuscript. It is much better to present results from WII and UKB together and I found the work flow description in Figure S3 very useful. I believe the main strength of the study is the consistency of clusters between WII and UKB. I also appreciate the response to my comment about the importance of reducing the data down from all combinations of the 4 conceptual dimensions for parsimony.

However, I'm still struggling to see the evidence that support the conclusion that RAR and chronotype dimensions alone don't fully capture circadian rhythm patterns – where is the evidence that the clusters identified better explain circadian rhythmicity or the total sleep-wake cycle? Can the authors assess performance of RAR and chronotype metrics alone in explaining circadian patterns compared with the nine derived clusters in the WII and UKB datasets, or demonstrate improved predictive validity of the clusters vs. RAR and chronotype alone for relevant health outcomes and/or behavioural traits? This would be required to justify the conclusions around “Better characterization of circadian rhythm profiles in real-world data allows their impact on adverse health events to be examined with greater accuracy”, which hasn't been directly assessed. While some health-related factors were investigated, these were treated as exposures and it would be of interest to investigate prospective associations with later disease/ill health as the outcome.

Title

I'm not convinced that this is an improvement on the previous one, in particular what is meant by “clusters of circadian rhythm”? Perhaps augment with “population clusters of circadian rhythm profiles” or something similar?

Abstract

Rephrase to remove double plurals, “Clusters participants differed on sociodemographic, behavioural and health-related factors” and “as demonstrated by the different dimensions combinations identified within clusters”

In addition, I'm not sure what is meant by – “Focussing only on RAR to measure CR might be an oversimplification, as demonstrated by the different dimensions combinations identified within clusters” ? Can you justify why it is a simplification to only focus on RAR.

Results

Every time a cluster is mentioned, please include in brackets the corresponding descriptives (e.g. RAR--/PA--/chronotype-) to aid interpretation

When describing associations, please state that the clusters are the dependent variables in the Results as well as the Methods.

Can p-values be added to Tables 3 and 4?

Please give exact p-values in the text rather than “was not statistically significant”

Was any consideration given to shift work patterns?

Discussion

Can some general patterns be identified between the cluster characteristics and sociodemographic/behavioural/health factors in the discussion?

Data on depressive symptoms and functional limitations (assessed through frailty index measures) are available in UK Biobank and so it's unclear why these have not been included.

(Remarks on code availability)

Reviewer #2

(Remarks to the Author)

The authors have thoroughly addressed all my comments, as well as those raised by others, to the best of my knowledge. With that said, I have no further remarks.

Kind regards,
Christian Benedict

(Remarks on code availability)

Reviewer #3

(Remarks to the Author)

I am happy with the authors response to my comments.

(Remarks on code availability)

Version 2:

Reviewer comments:

Reviewer #1

(Remarks to the Author)

I appreciate the authors' thorough revisions and thoughtful responses to the previous comments. All of my concerns have been adequately addressed, and the manuscript has been significantly improved. I have no further suggestions.

(Remarks on code availability)

Code for both the UK Biobank and Whitehall studies have been included, with an accompanying README file. Without access to the raw data, I am unable to run the code but it appears comprehensive.

RESPONSE TO REVIEWERS

REVIEWER 1 COMMENTS

GENERAL COMMENT: This study uses a clustering approach to identify nine distinct circadian rhythm phenotypes based on accelerometer-based metrics from the Whitehall II study (n=3,991) and UK Biobank (n=54,995). These phenotypes are categorized by combinations of rest-activity rhythm (RAR), daytime activity, sleep efficiency and chronotype metrics. The derived phenotypes were found to be associated with a number of demographic, behavioural and health traits including age, smoking, light exposure, BMI, depressive symptoms and medication use. The work appears original in its methodology of combining four dimensions of circadian rhythm and identifying a variety of phenotypes. However, the proposed value and significance of identifying the new multi-faceted phenotypes is unclear. I was not convinced that the work would be of sufficient interest or significance to the field, and proposed links between the phenotypes and health outcomes was not clearly demonstrated. More specific comments and questions are outlined below

OUR RESPONSE: We thank the reviewer for highlighting the originality of our approach and we hope that the revised manuscript addresses the reviewer's concerns. We have removed the term 'phenotype' to avoid confusion, and focus on the method used to derive clusters of circadian rhythm. The significance of our research is to show that the multi-faceted aspects of rest-activity rhythm, physical activity, sleep, and chronotype can be used together to characterise circadian rhythm. Due to complexity of treating these data previous studies have mainly focussed on rest-activity rhythm, that is not informative of the total sleep-wake cycle. Our primary objective was to demonstrate the validity of this new approach in two independent cohort studies. As highlighted by reviewer 3, we observed expected associations with the different socio-demographic, behavioural and health-related factors, reinforcing the construct validity of our approach.

ABSTRACT

1. The method/methods used to identify circadian rhythm phenotypes are not described

OUR RESPONSE: Thank you, we have revised the abstract and added the following information:

"We propose a two-step method: 1. data reduction of CR metrics using principal component analyses, 2. k-means clustering to identify groups of individuals with the same profile."

2. What is meant by 'opposite RAR and daytime activity'?

OUR RESPONSE: Apologies, the sentence has been revised as follows to better describe the clusters.

"Nine CR clusters were identified: two presented extreme (most robust/poorest) RAR and (highest/lowest) daytime activity, two robust RAR with opposite sleep profiles (longer and efficient/shorter and fragmented), one high-intensity physical activity, and four poor RAR (one characterized by late chronotype, two by low physical activity but opposite sleep profiles, and one by restless (agitated) sleep)."

3. What is the distinction between inefficient and restless sleep?

OUR RESPONSE: Inefficient sleep refers to disturbed sleep based on measures of sleep duration, efficiency and fragmentation while restless sleep corresponds to agitated sleep (mean acceleration

during the night above 1SD from the mean). We have added this clarification to the abstract (see response above) and to the results section.

Results, page 5, paragraph 3:

“Participants in cluster 8 (RAR-/PA+/Restless sleep) showed restless (agitated) sleep where mean acceleration during sleep was the highest (> 1SD and 2SD from the sample mean in WII and UKB, respectively).”

4. What is value of identifying composite phenotypes rather than the 4 conceptual dimension and how interpretable/meaningful are the associations identified?

OUR RESPONSE: Thank you for this insightful comment. Theoretically, there is another way of analysing these data. Using the four dimensions (rest-activity rhythm, physical activity, sleep, and chronotype) an individual can be classified as having a poor, intermediate, or good profile. All possible combinations of these dimensions would lead to 3⁴ or 81 combinations. While this approach is feasible, it has little practical value for research. Furthermore, the correlation between measures of these dimensions implies that many of the 81 possible combinations would not be seen in the general population. Our 2-step approach is parsimonious, and nine clusters of circadian rhythm were found in two independent cohorts. The value of identifying composite clusters instead of studying the 4 conceptual dimensions separately is that they represent real-life human circadian rhythm profiles.

INTRODUCTION

5. Explain distinction (and correlation) between the 4 different conceptual dimensions

OUR RESPONSE: We have revised the introduction to include a description of the four conceptual dimensions. Please note that these dimensions are measured using multiple measures and their correlations are shown in Figure 1.

Introduction, page 3, paragraph 2-3:

“Multiple metrics can be derived from 24-hour accelerometer data to reflect four key behavioural dimensions of circadian rhythm – ‘rest-activity rhythm’ (RAR) reflecting circadian rhythmicity in a free-living setting, ‘daytime activity’ composed of physical activity (PA) and sedentary behaviour (SB) over the waking period, ‘sleep’ to measure the quality and quantity of sleep during the sleep period, and ‘chronotype’ to measure wakefulness and sleep timing. Whether dysregulation is uniform across these circadian rhythm dimensions in individuals remains unclear. (...)

The quantity of data generated by accelerometers, with multiple metrics for each dimension, and their correlated nature present analytic challenges.”

6. Can accelerometers identify true chronotype as defined by ‘natural preference’?

OUR RESPONSE: We thank the reviewer for this interesting comment. Accelerometers provide data over a 24-h cycle for several days and 5 chronotype metrics are derived from these data (see Table S1 for the description of each of the 36 metrics). We have added to the description of chronotype in the manuscript to be more specific: *“‘chronotype’ to measure wakefulness and sleep timing”*.

‘Natural preference’ cannot be measured by accelerometer but in UK Biobank there were questions on preference. We undertook further analysis (shown in Supplementary Table S7) suggested by reviewer 2 (comment 3) to show that participants with self-reported circadian preference of “Definitely an evening person” were often in clusters 5 and 9, characterized as late chronotype in our analysis of the accelerometer data.

RESULTS

7. The range of correlations between metrics should be included in the text.

OUR RESPONSE: Thank you, we added the range of correlations between RAR and the other dimensions in the revised manuscript as suggested by the reviewer.

Results, page 4, paragraph 2:

*“The correlations (**Figure 1**) between the 36 accelerometer-based metrics (described in **Table S1**) were mostly moderate (0.40 to 0.59) to high (≥ 0.60) for metrics within the same dimension of circadian rhythm - RAR, daytime activity, sleep, and chronotype. RAR metrics were also highly correlated with most daytime activity metrics (e.g. absolute range of correlations of cosinor mesor with 13 out of 15 daytime activity metrics: 0.53 to 0.86 in WII, 0.41 to 0.83 in UKB), moderately correlated with some sleep metrics (absolute range of correlations of relative amplitude with 6 out of 10 sleep metrics: 0.27-0.57 in WII, 0.17-0.76 in UKB), and had a weak correlation with chronotype metrics (absolute range of correlations for the 5 metrics: 0.11-0.29 in WII, 0.11-0.31 in UKB). The correlations between metrics of daytime activity, sleep, and chronotype were weaker (< 0.40).”*

8. Some terms were not defined in the Results and would aid interpretation e.g. IG, mesor, relative amplitude

OUR RESPONSE: Thank you. We have indicated in the results section that **Table S1** contains a full description of the metrics.

Results, page 4, paragraph 2:

*“The correlations (**Figure 1**) between the 36 accelerometer-based metrics (described in **Table S1**)...”*

9. When were the characteristics assessed with respect to accelerometer wear?

OUR RESPONSE: This information is provided in the method section as follows:

Methods, page 14, paragraph 5:

“In WII, covariates were measured at the 2012-2013 wave either using questionnaire, clinical examination, or data from electronic health records, including the Hospital Episode Statistics (HES) and Mental Health Services Data Set. In UKB, covariates were either extracted from the baseline examination in 2006-2010 or extracted from electronic health records (HES).”

We have also added this information in the revised results section.

Results, page 6, paragraph 3:

*“Independent associations of sociodemographic, behavioural, and health-related factors with circadian rhythm clusters were examined using multinomial regressions among participants with complete data (N=3,965 in WII and N=51,507 in UKB). These covariates were measured concurrently to accelerometer measures except, in UKB, for marital status, employment status, Townsend deprivation index, behavioural factors, body mass index (BMI), hypertension, hyperlipidaemia, and central nervous system (CNS) medication, that were measured in 2006-2010 (in average (SD) 5.7 (1.1) years before accelerometer measure). **Tables 1** (WII) and **2** (UKB) show participants’ characteristics in the nine circadian rhythm clusters.”*

10. It is difficult to get sense of magnitude/statistical robustness for associations between characteristics and CR phenotypes, hindering interpretation

OUR RESPONSE: Tables 3 and 4 provide results showing the magnitude of these associations, similar findings in both cohort studies also suggests that the results are robust. In addition, we completely revised the description of the results section to describe findings in WII and UK Biobank together. We took care to describe both magnitude and gradient of association with the different clusters. Please see the revised section “Factors associated with circadian rhythm clusters” in the Results section (pages 6-8).

11. Inference from season difficult as presumably the date the participants were given accelerometer devices were largely random, and there were no repeated measures between seasons?

OUR RESPONSE: We agree with the reviewer, and in the revised manuscript we chose to remove the association with Season from the results although it was retained as a covariate as in most studies on accelerometers to account for seasonal variability (temperature, light exposure, and behavior) that might affect findings.

12. Mention 36 metrics for each cluster but only RAR, PA, SB, LIPA, Sleep, late chronotype, restless sleep described - why? How do these map to the 4 conceptual dimensions?

OUR RESPONSE: Apologies for the confusion. We have extensively revised the manuscript to better describe the two-step procedure, shown in Supplementary **Figure S3** and copied below. Table S1 contains the description of all 36 metrics used to measure the four dimensions - RAR is measured by 6 metrics, daytime activity by 15 metrics, sleep by 10 metrics, and chronotype by 5 metrics. The two-step procedure (shown below) involved data reduction (using principal components analysis) and then cluster analysis to identify groups of participants with similar pattern of circadian rhythm (the clusters). The cluster labels (heading in Tables 3 and 4) were chosen to reflect the predominant features of the given cluster.

13. Describe in more detail the cluster based analysis to get from 8 PCs to 9 clusters/phenotypes
OUR RESPONSE: Thank you for this feedback. The manuscript has been carefully revised to better explain our analytic strategy. We provide a step-by-step description of the methods leading from 36 metrics, to 8 principal components, and 9 clusters in **Supplementary methods** and **Supplementary Figure S3** (shown in response to comment 12 above). We also provide further information on these steps in **Supplementary Table S2** and **Figures S4** and **S5**.

14. I'm struggling to understand the utility/purpose of classifying these composite phenotypes rather than grouping based on conceptual dimensions. The authors mention that use of conceptual dimensions may not classify as accurately, but it would be useful to demonstrate/quantify this

OUR RESPONSE: Please see our response to comment 4.

Our approach allows all dimensions of circadian rhythm (rest-activity rhythm, daytime activity, sleep, and chronotype) to be used in the construction of the clusters. The introduction has been revised to reflect our approach.

Introduction, page 3, paragraph 2-3:

“Multiple metrics can be derived from 24-hour accelerometer data to reflect four key behavioural dimensions of circadian rhythm – ‘rest-activity rhythm’ (RAR) reflecting circadian rhythmicity in a free-living setting, ‘daytime activity’ composed of physical activity (PA) and sedentary behaviour (SB) over the waking period, ‘sleep’ to measure the quality and quantity of sleep during the sleep period, and ‘chronotype’ to measure wakefulness and sleep timing. Whether dysregulation is uniform across these circadian rhythm dimensions in individuals remains unclear. Most studies on accelerometer-based measures of circadian rhythm focus on RAR and chronotype, either ignoring other dimensions⁶⁻⁸, or one marker of PA and/or sleep.⁹⁻¹² Few studies have examined all four dimensions (RAR, daytime activity, sleep, and chronotype)¹³⁻¹⁵ and either considered these dimensions in separate models¹³, or in a mutually adjusted model¹⁴, or derived a large number of composite scores,¹⁵ that do not allow individual patterns of dysregulations in the four dimensions to be elucidated.

The quantity of data generated by accelerometers, with multiple metrics for each dimension, and their correlated nature present analytic challenges. The aim of this study was to consider all four dimensions (RAR, daytime activity, sleep, and chronotype) and their metrics to identify real-life human clusters of circadian rhythm in older adults.”

15. The results of replication in UKB quite poorly described

OUR RESPONSE: We apologise. Comments from reviewers led us to rerun the analyses in both cohort studies in parallel and describe the results together. Please see the revised Results section.

16. More description of e.g. proportion of people within the different phenotypic groups in the different populations would be useful

OUR RESPONSE: Thank you, we added the % in the table of characteristics (Tables 1 and 2) and when showing results of associations with covariates (Tables 3 and 4) and refer to these proportions in the results and discussion sections.

Results, page 6, paragraph 2:

“Clusters 3 and 6 in both cohort studies were the largest (15.7/17.1% (WII/UKB) and 17.7/16.1%, respectively) and clusters 8 and 9 the smallest (5.9/3.4% and 6.5/6.9%, respectively). The size of clusters was similar in WII and UKB, with the largest difference observed for the size of cluster 4 (9.5 and 13.0%).”

Discussion, page 10, paragraph 2:

“In our study the size of the nine circadian rhythm clusters was slightly different in WII and UKB, possibly due to difference in the distribution of age, sex, and education as well as differences in accelerometer protocol.”

17. Were/can the clusters from WHII be directly overlayed into UKB and was goodness of fit assessed?

OUR RESPONSE: As described in the methods section the intention was to assess whether the cluster structure (grouping participants according to circadian rhythm dimensions) was similar in two independent cohort studies. The analyses to derive clusters were conducted separately, and as shown in Figures 2 and 3, the identified clusters were fairly similar. The differences were small and are described in the Results section (see below).

The number of clusters were determined using statistical criteria; the procedure is described in detail in Supplementary Methods: Identification of circadian rhythm clusters. Findings for these criteria are shown in Supplementary table S2 and Figures S4 and S5.

Results, page 5, paragraph 2-3:

“Participants in clusters 2 and 3 had robust (+) RAR but opposite sleep profiles with cluster 2 characterised by shorter sleep duration and more fragmented sleep (Sleep-) and cluster 3 by longer sleep duration and efficient sleep (Sleep+). In addition, in WII, cluster 2 was also characterized by more LIPA (LIPA+) and in UKB by both more LIPA and MVPA (PA+).

(...)

Participants in clusters 5, 6, 7, and 8 had poor (-) RAR pattern. Cluster 5 (RAR-/Chronotype--) had the most delayed (--) chronotype with later sleep period and later activity during the day. Participants in cluster 6 (RAR-/PA-/Sleep+) and 7 (RAR-/PA-/Sleep--) had both low daytime activity but differed in sleep. Participants in cluster 8 (RAR-/PA+/Restless sleep) showed restless (agitated) sleep where mean acceleration during sleep was the highest (> 1SD and 2SD from the sample mean in WII and UKB, respectively. In WII, participants in cluster 8 also presented shorter and less efficient sleep.”

18. Mention is made to differences in some phenotype differences e.g. fruit and veg consumption in Discussion but not in Results

OUR RESPONSE: Thank you; we added the information in the results section of the revised manuscript.

Results, page 7, paragraph 2:

“Individuals who consumed less fruit and vegetables were more likely to be in clusters 7 (RAR-/PA-/Sleep-) and 9 (RAR--/PA--/(Chronotype- in UKB)), and in UKB also in clusters 5 (RAR-/Chronotype--) and 6 (RAR-/PA-/Sleep+).”

DISCUSSION

19. The authors state “As some covariates were not available in the UK Biobank study, we could not undertake a strict replication of the analysis on associated factors” - is this referring to characteristics and/or sleep metrics? A better attempt at assessing consistency in associations between the studies could be made for those common variables that do exist

OUR RESPONSE: We thank the reviewer for this comment and apologise for the lack of clarity. This sentence referred to lack of data on covariates - depressive symptoms and functional limitations in UK Biobank. In the revised manuscript we chose to focus on covariates available in both cohort studies in the main results. Findings for covariates available only in the Whitehall study are described in sensitivity analysis (page 8, paragraph 2).

METHODS

20. The authors describe how the cluster closest to the population mean is used as the reference category for multinomial regression in methods - how was this determined?

OUR RESPONSE: We have better described the procedure in the revised manuscript, as follows:

Methods, page 16, paragraph 4:

“The reference category in the multinomial regression was chosen based on size (number of participants), the cluster where none of the metric means was above or below one SD from the population mean, and among the retained, the healthier was selected.”

21. What was the exposure and what was outcome in the association analysis with characteristics - this should be more clearly articulated.

OUR RESPONSE: Thank you, we revised the sentence as below.

Methods, page 16, paragraph 4:

“Once circadian rhythm clusters were identified, we used multinomial regression to examine associations of participants’ characteristics (independent variables) and season of wear with circadian rhythm clusters (dependent variable).”

22. In multinomial regression, were all variables mutually adjusted and can this be justified?

OUR RESPONSE: The primary focus of the manuscript was to establish a robust method to allow use of complex accelerometer data on the entire sleep-wake cycle. The analyses with covariates were from a mutually adjusted model to show the independent associations between each covariate and the clusters. We have added the following text to the manuscript.

Results, page 6, paragraph 3:

“Independent associations of sociodemographic, behavioural, and health-related factors with circadian rhythm clusters were examined using multinomial regressions among participants with complete data (N=3,965 in WII and N=51,507 in UKB).”

23. Methods for reporting UKB are too brief - ‘we repeated analyses with some modifications’

OUR RESPONSE: We agree. Comments from reviewers led us to rerun the analyses in parallel so that the results could be described together. These changes are shown in the manuscript as below.

Methods, page 15, paragraph 4:

“All analyses were conducted in parallel in the two cohort studies to identify how circadian rhythm metrics clustered in individuals, and then examine the association of the clusters with covariates.”

24. The decision and justification of restricting UKB individuals to only those over age of 60 needs to be explained

OUR RESPONSE: Apologies for the lack of clarity. The revised manuscript now carries a justification. Our focus was on older adults where circadian rhythm dysregulations are common.

Introduction, page 3, paragraph 3:

“The aim of this study was to consider all four dimensions (RAR, daytime activity, sleep, and chronotype) and their metrics to identify real-life human clusters of circadian rhythm in older adults.”

25. It is unclear why Whitehall II is treated as the primary analysis when it is >10x smaller than UKB

OUR RESPONSE: We thank the reviewer for this comment. In response to this comment and several other comments, we revised our approach by undertaking the analyses in parallel in the two cohort studies to test the robustness of the findings. We show that, despite several differences between the cohort studies—sociodemographic factors, time period, wear protocol (use or not of a sleep diary, different devices used, the wrist on which the accelerometer was worn, differences in recording frequency)—the resulting circadian rhythm clusters were similar.

26. Authors state that code will be shared on GitHub upon publication of the paper so I have been unable to review this.

OUR RESPONSE: We will post the code on GitHub when the paper will be published. For now, we send all the codes as supplementary file.

REVIEWER 2 COMMENTS

1. I believe this work has significant merit.

OUR RESPONSE: We thank the reviewer for this positive feedback.

2. Considering the associations in Tables 2 and 3, the relationships with the nine circadian rhythm phenotypes illustrate how both extrinsic and intrinsic factors shape the timing and intensity of behaviours across 24 hours. To what extent are certain circadian rhythm phenotypes primarily driven by external factors? Do they represent true circadian phenotypes (i.e., behaviours that persist under constant routine conditions), or do they simply reflect discrepancies between an individual's preferred behavioural timing and the constraints imposed by life circumstances (e.g., parenting responsibilities, work schedules, or leisure activity timing)?

OUR RESPONSE: This is an excellent comment, and we thank the reviewer. As described in the revised introduction, we see the sleep-wake cycle as the bio-behavioural manifestation of an internal biological clock. This manifestation is bound to be guided by both intrinsic and external factors, both influencing the behavioural expression of the circadian rhythm. Our focus was on elaboration of a method that would allow all measurable dimensions over a 24-h cycle (rest-activity rhythm (RAR), daytime activity, sleep, and chronotype) to be used for a parsimonious, yet comprehensive measure of the behavioural expression of circadian rhythm. Please see the discussion for further elaboration of the role of extrinsic and intrinsic factors.

Discussion, page 10, paragraph 1:

“Our two-step approach (PCA for data reduction followed by cluster analyses) captures the behavioural manifestation of the endogenous biological clock by capturing multiple aspects of the sleep-wake cycle in older adults using data from two population-based cohort studies. Accelerometers provide scalable, cost-effective measures of the sleep-wake cycle - one of the most visible manifestations of the circadian clock. It remains important to consider the influence of extrinsic factors, such as seasonal variations, or activity routines that may also influence individual behaviours. For example, we found employment status to affect the clusters identified in our study, as also demonstrated in another study where work environment and schedule impacted sleep.²⁷ Exposure to zeitgebers such as light plays an important role in the synchronisation of the internal clock, and misalignment of zeitgebers with endogenous biological clock can lead to chronic perturbations of circadian rhythm.²⁸ Further studies are needed to determine the association between behavioural manifestation of the biological clock and its molecular and genetic markers.”

3. If I'm not mistaken, participants—at least in the UK Biobank—were asked about their circadian preference (chronotype). How prevalent are the subjective chronotype subtypes within each of your nine identified phenotypes?

OUR RESPONSE: We thank the reviewer for this suggestion. In the revised manuscript we report results of sensitivity analysis on the prevalence of subjective chronotype variable in the circadian rhythm clusters (Table S7).

Results section, page 6, paragraph 2:

“Data on subjective chronotype preference in UKB (Table S7) showed 8.2% of participants to report themselves as being “Definitely an evening person” and this was particularly the case for participants in clusters identified with delayed chronotype using accelerometer data - clusters 5 (RAR-/Chronotype--)

and 9 (RAR--/PA--/Chronotype-) (21.9% and 16.1%, respectively, compared to 6.0% in cluster 3, $p < 0.001$).”

4. You found that daily light exposure had a strong association with the odds of being assigned to a certain circadian phenotype. Did you exclude individuals with blindness (e.g., see Data-Field 131212)?

OUR RESPONSE: To respond to this comment we examined the number of blind participants - 3 cases of blindness in WII (<0.01%) and 172 in UK Biobank (<0.01%). The small numbers are unlikely to influence the association between light exposure and circadian rhythm clusters. We repeated the analyses excluding these cases in UKB, the results were unchanged (see table below).

As the manuscript is already quite long, we only make a brief mention of this aspect in the Discussion section “As the number of blind participants was small (<0.01% in WII and UKB), this group is unlikely to influence findings on light exposure”.

	Cluster 1	Cluster 2	Cluster 3	Cluster 4	Cluster 5	Cluster 6	Cluster 7	Cluster 8	Cluster 9
	RAR ++ PA ++	RAR + PA + Sleep -	RAR + LIPA + Sleep +	MVPA ++	RAR - Chronotype --	RAR - PA - Sleep +	RAR - PA - Sleep --	RAR - PA + Restless sleep	RAR -- PA -- Chronotype -
Original findings, N=51,507									
Daily % >1000 lux, 1SD higher exposure	1.26 (1.21-1.31)	1.08 (1.04-1.12)	Ref	0.99 (0.96-1.03)	0.81 (0.78-0.85)	0.82 (0.79-0.85)	0.87 (0.83-0.91)	0.94 (0.88-1.00)	0.69 (0.65-0.72)
Findings excluding blind participants, N=51,335									
Daily % >1000 lux, 1SD higher exposure	1.26 (1.21-1.31)	1.08 (1.04-1.12)	Ref	0.99 (0.95-1.03)	0.82 (0.79-0.85)	0.82 (0.79-0.85)	0.87 (0.83-0.90)	0.94 (0.89-0.99)	0.68 (0.65-0.72)

LIPA, light intensity physical activity; MVPA, moderate to vigorous physical activity; PA, physical activity; RAR, rest-activity rhythm; SD, standard deviation.

Bold values indicate significance at $P < 0.05$.

5. Given the association between both age and female sex with the circadian phenotypes, I wonder if there is an interaction between female sex and age.

OUR RESPONSE: We thank the reviewer for this suggestion, we examined the interaction between age and sex and did not find an interaction (p -value for log-likelihood ratio test: 0.40 and 1.00 in WII and UKB, respectively).

6. Why was napping not considered in the construction of your circadian phenotypes?

OUR RESPONSE: This is an interesting point, thank you. Most research on sleep is on nocturnal sleep and few studies exist on the detection of naps using accelerometers where the challenge is to differentiate naps from sedentary time. Simpler devices (Actigraph (Actiwatch (Kanadi et al, J Sleep Res, 2011), ActiSleep (Liu et Benjamin-Neelson, Sleep, 2023)) have derived data on naps using the Actiwatch software. However, accelerometers (GeneActiv, Axivity) produce raw acceleration that does not allow use of the Actisleep software.

In children naps can be measured with some accuracy (Liu et Benjamin-Neelson, Sleep, 2023) as they are likely to be asleep when not moving. In older adults where a high proportion of daytime is spent in sedentary behaviour, the challenge is to differentiate between sedentary time (for example, watching TV) and napping. Some attempts have been made using the same algorithm as the one to detect the sleep periods during the night. In a previous study based in UK Biobank, the number of naps was computed as the number of sleep periods (no rotation of the wrist of more than 5° over a 5-minute moving window) lasting at least 30 minutes outside the primary sleep period (Wainberg et al, PLoS Med, 2021). In another study (Jones et al, Nature communications, 2019), any 5 minutes detected as no rotation of the wrist of more than 5° was classified as “diurnal inactivity capturing very inactive states such as napping and wakeful rest”. Further research is needed to determine the validity of these approaches.

The lack of consensus of measuring naps using accelerometry led us we chose not to isolate these measures in our analyses. Please note that naps are likely to contribute to measures on the fragmentation of the circadian rhythm such as interdaily variability index and the transition probability from activity to rest.

We have added the following sentence in the limitation section of the discussion:

Discussion, page 11, paragraph 2:

“Fourth, in the absence of a consensus on measuring naps, we did not use data on naps in the identification of circadian rhythm clusters. We recognize that naps are likely to contribute to intradaily variability, transition probabilities from activity to rest, and sedentary behaviour.”

References:

- Kanady, J. C., Drummond, S. P. & Mednick, S. C. Actigraphic assessment of a polysomnographic-recorded nap: a validation study. *J Sleep Res* **20**, 214-222, doi:10.1111/j.1365-2869.2010.00858.x (2011).
- Liu, T. & Benjamin-Neelson, S. E. A longitudinal study of infant 24-hour sleep: comparisons of sleep diary and accelerometer with different algorithms. *Sleep* **46**, doi:10.1093/sleep/zsad160 (2023).
- Wainberg, M. et al. Association of accelerometer-derived sleep measures with lifetime psychiatric diagnoses: A cross-sectional study of 89,205 participants from the UK Biobank. *PLoS Med* **18**, e1003782, doi:10.1371/journal.pmed.1003782 (2021).
- Jones, S. E. et al. Genetic studies of accelerometer-based sleep measures yield new insights into human sleep behaviour. *Nat Commun* **10**, 1585, doi:10.1038/s41467-019-09576-1 (2019).

7. Minor correction: In Table 2, you stated that the reference group consisted of 625 subjects.

OUR RESPONSE: Apologies. The cluster derivation was based on maximum numbers but the “association analysis” was based on available data on covariates. The flowchart (Supplementary Figure S1 and S2) and the Results section clarify these differences in N.

Results, page 4, paragraph 2:

*“The clustering analyses were based on 3,991 (mean age=69.4 (standard deviation (SD))=5.7, range=60-83) years; 26% women) and 54,995 (mean age=67.5 (SD)=4.2, range=60-79) years; 54% women) individuals in WII and UKB, respectively (flowcharts in **Figures S1 and S2**).”*

Results, page 6, paragraph 3:

“Independent associations of sociodemographic, behavioural, and health-related factors with circadian rhythm clusters were examined using multinomial regressions among participants with complete data (N=3,965 in WII and N=51,507 in UKB).”

8. I've checked the prevalence of the various circadian phenotypes in the Whitehall Study and UK Biobank study. They differ somewhat. However, both cohorts are from the UK, and the data were collected at different times, correct? Does this suggest that not only seasons are associated with certain circadian phenotypes, but also that longer biological time scales may have an impact?

OUR RESPONSE: We thank the reviewer for this comment. The prevalence of circadian rhythm clusters (now provided in Tables 1, 2, 3, and 4) varies moderately, and there could be a number of reasons – mentioned in the Results and Discussion section of the revised manuscript.

Results, page 6, paragraph 2:

“Clusters 3 and 6 in both cohort studies were the largest (15.7/17.1% (WII/UKB) and 17.7/16.1%, respectively) and clusters 8 and 9 the smallest (5.9/3.4% and 6.5/6.9%, respectively). The size of clusters was similar in WII and UKB, with the largest difference observed for the size of cluster 4 (9.5 and 13.0%).”

Discussion, page 10, paragraph 2:

“In our study the size of the nine circadian rhythm clusters was slightly different in WII and UKB, possibly due to difference in the distribution of age, sex, and education as well as differences in accelerometer protocol.”

9. Given the potential impact of extrinsic factors (e.g., noise), I wonder if you could expand your socio-demographic analysis to include the area of residence.

OUR RESPONSE: Thank you for this excellent suggestion. In the revised analyses we included an area deprivation index in both studies (Townsend in UKB and index of multiple deprivation in WII) to consider these extrinsic factors. Findings are shown in the tables 1 and 2 (characteristics), and 3 and 4 (associations analyses) and described in the results and discussion section.

Results, page 7, paragraph 1:

“WII participants living in more deprived areas were more likely to be in clusters characterized by disturbed sleep and late chronotype (clusters 2, 5, 7, and 8) and in UKB they were more likely to be in the cluster with high-intensity PA (cluster 4) and all clusters characterised by poorer RAR (clusters 5 to 9).”

Discussion, page 11, paragraph 1:

“Living in a more deprived area was more common in clusters characterized by sleep disturbances and delayed chronotype in WII, whereas in UKB it was more common in clusters with poorer RAR.”

REVIEWER 3 COMMENTS

1. This is a nicely written study with some interesting data

OUR RESPONSE: We thank the reviewer for this positive feedback.

2. My major concern is that I am not clear how the authors have demonstrated that these 9 metrics are measuring circadian phenotypes. PCA and clustering analyses are performed but it not clear why the authors think these are measuring specifically circadian phenotypes. The fact that the clusters appear to be similarly recreated in UK Biobank is encouraging - suggesting it is not just an artefact of clustering. But why the authors are sure that these 9 phenotypes represent new circadian features needs to be made clearer to me.

OUR RESPONSE: Thank you for this comment that led us to revise the introduction and remove the term "phenotype". As reflected in the revised introduction, our aim was to generate measure of the sleep-wake-cycle, a bio-behavioural manifestation of circadian rhythm. The nine clusters we identified do not

aim to be new circadian rhythm features. Instead, the challenge was to use all the data (36 metrics to reflect 4 dimensions (RAR, daytime activity, sleep, and chronotype)) from accelerometers on the sleep wake-cycle over 24h to construct clusters that identify differences in circadian rhythm. Most studies on circadian rhythm use one or two of these dimensions as using all 36 metrics together presents several challenges. Our study provides a solution to these challenges, and similar results in two studies shows the robustness of findings.

3. Correlation analyses against a range of phenotypes is performed and the new circadian phenotypes appear to associate with what might be expected. It is difficult to make to much of these epidemiological analyses though and I wonder if genetics could be used to identify whether these seem to be reasonable clusters as has been done for other diseases (e.g. <https://link.springer.com/article/10.1007/s00125-022-05848-6>).

OUR RESPONSE: We might not have understood this comment but our intention was not to map genetic factor clusters associated with individual diseases on circadian rhythm.

The paper recommended by the reviewer proposes a method to identify clusters of genetic loci for type 2 diabetes, and then for coronary artery disease and chronic kidney disease. The authors of that paper write *“we have identified ten robust genetic clusters pointing to mechanistic pathways of type 2 diabetes using a high-throughput clustering pipeline”*.

Our manuscript does not deal with a “highly polygenic disease” such as type 2 diabetes where multiple GWAS datasets exist.

Our focus was to capture variations in measures of the sleep-wake cycle – a bio-behavioural measure that reflects circadian rhythm. To use the same approach as recommended by the reviewer, our focus would need to shift to circadian rhythm disorder (ICD-10 G47.23) and we would need GWAS for accelerometer-based measures of circadian rhythm disorder. These do not exist.

Four studies have examined the genetic variants related to some accelerometer measures (see table at the end of this document). Out of the 36 metrics we used, a GWAS was conducted only for 9 of them. For sleep duration and overall activity (or equivalent mean acceleration), the SNPs identified differ from one study to the other (from the 58 identified SNPs, none of them appears simultaneously in the four studies, neither in the three studies, and only three SNPs appear in two studies). Given these concerns, we do not see a way to use the method proposed by the reviewer.

4. I have an additional concern that, at least for the UK Biobank, activity monitors were worn some times several years after the baseline measurements. Does this affect interpretation of any of the association results?

OUR RESPONSE: We agree, it is a limitation for the association analyses in UK Biobank. However, it does not affect findings on derivation of circadian rhythm clusters for which baseline measurements are not used. The difference in measurement of covariates is referred to in the discussion section.

Discussion, page 11, paragraph 2:

“Three, in UKB some covariates were measured at inclusion rather than concurrently with the accelerometer measure but a previous study in UKB did not find major differences in associations for covariates drawn from baseline or at the same time as the accelerometer.³³ In addition, the pattern of results in our study was similar in WHI and UKB, suggesting little impact of when covariates were measured.”

Table R1. Summary of GWAS on accelerometer-assessed circadian rhythm

Trait	SNP	Risk allele/ other allele	P	β
Jones et al., Nature Communications , 2019				
L5 timing	rs1144566	C/T	8E-12	0.096
	rs113851554	T/G	2E-35	0.133
	rs12991815	C/G	2E-09	0.029
	rs9369062	A/C	9E-14	0.039
	rs4882315	T/C	2E-08	0.027
	rs12927162	G/A	3E-08	0.029
M10 timing	rs1973293	C/T	1E-09	0.029
Sleep midpoint	rs11892220	T/A	3E-08	0.029
Sleep duration	rs2660302	A/T	9E-12	0.041
	rs113851554	G/T	2E-25	0.110
	rs62158170	G/A	3E-21	0.054
	rs17400325	T/C	2E-08	0.066
	rs72828540	T/C	1E-13	0.041
	rs9369062	C/A	2E-10	0.033
	rs2975734	C/G	1E-08	0.027
	rs13282541	C/T	4E-09	0.032
	rs2880370	A/T	2E-08	0.028
	rs800165	C/T	3E-08	0.028
	rs10138240	G/C	7E-10	0.029
Sleep efficiency	rs113851554	G/T	5E-22	0.101
	rs62158169	T/C	2E-08	0.032
	rs17400325	T/C	2E-10	0.074
	rs13094687	G/A	1E-08	0.029
	rs13080973	G/A	3E-08	0.032
N sleep bouts	rs12714404	T/G	1E-12	0.037
	rs310727	T/C	3E-08	0.026
	rs55754932	C/A	2E-12	0.037
	rs9864672	T/C	2E-10	0.029
	rs4974697	T/A	5E-08	0.026
	rs7377083	A/C	2E-09	0.029
	rs749100	A/G	9E-12	0.033
	rs9341399	C/T	6E-12	0.066
	rs1889978	C/T	5E-09	0.027

Table R1 continued. Summary of GWAS on accelerometer-assessed circadian rhythm

Trait	SNP	Risk allele/ other allele	P	β
N sleep bouts	rs2141277	A/G	1E-08	0.026
	rs10233848	G/A	2E-11	0.035
	rs1124116	A/G	2E-09	0.031
	rs4755731	G/A	3E-09	0.028
	rs3751837	C/T	4E-09	0.033
	rs8045740	G/T	6E-14	0.052
	rs11078917	A/C	3E-08	0.029
	rs11082030	T/C	8E-09	0.030
	rs8098424	G/A	1E-08	0.027
	rs76753486	T/C	2E-08	0.047
	rs429358	T/C	4E-08	0.036
rs12479469	A/G	4E-10	0.031	
Doherty et al., Nature Communications . 2018				
Overall activity	rs564819152	A/G	4.20E-09	0.028
	rs2696625	A/G	3.20E-12	-0.037
	rs59499656	A/T	1.90E-09	-0.028
Sedentary	rs26579	G/C	2.60E-09	0.028
	rs25981	G/C	3.00E-09	0.028
	rs1858242	A/G	3.10E-09	0.031
	rs34858520	A/G	4.20E-09	0.028
Sleep duration	rs2416963	C/T	2.30E-10	0.030
	rs2006810	T/C	3.90E-09	-0.028
	rs62158170	A/G	5.80E-20	-0.051
	rs113851554	G/T	3.10E-18	0.090
	rs72828533	A/T	2.70E-13	0.043
	rs7502280	T/G	8.80E-11	0.051
	rs2303100	C/T	1.40E-10	-0.030
	rs75641275	A/C	2.20E-10	0.042
Klimentidis et al., Int J Obes . 2018				
Mean acceleration	rs55657917	T	5.0E-12	-0.30*
	rs59499656	A	2.4E-09	-0.23*
Fraction of acceleration >425mg	rs743580	A	1.3E-09	0.025*
Ramadan et al., BMJ Open Sp Ex Med , 2022				
MVPA fraction of accelerations >100mg	rs4480415	G/A	4.6e-08	-0.012
	rs17443704	G/A	3.5e-08	-0.013

L5, least active 5-hour period; M10, most active 10-hour period; MVPA, moderate to vigorous physical activity; VPA, vigorous physical activity

* β /OR (odds ratio).

References:

- Jones, S. E. *et al.* Genetic studies of accelerometer-based sleep measures yield new insights into human sleep behaviour. *Nat Commun* **10**, 1585, doi:10.1038/s41467-019-09576-1 (2019).
- Doherty, A. *et al.* GWAS identifies 14 loci for device-measured physical activity and sleep duration. *Nat Commun* **9**, 5257, doi:10.1038/s41467-018-07743-4 (2018).
- Klimentidis, Y. C. *et al.* Genome-wide association study of habitual physical activity in over 377,000 UK Biobank participants identifies multiple variants including CADM2 and APOE. *Int J Obes (Lond)* **42**, 1161-1176, doi:10.1038/s41366-018-0120-3 (2018).
- Ramadan, F. A. *et al.* Association of sedentary and physical activity behaviours with body composition: a genome-wide association and Mendelian randomisation study. *BMJ Open Sport Exerc Med* **8**, e001291, doi:10.1136/bmjsem-2021-001291 (2022).

Reviewer #1

COMMENT 1. The authors have made a strong attempt to improve the manuscript. It is much better to present results from WII and UKB together and I found the work flow description in Figure S3 very useful. I believe the main strength of the study is the consistency of clusters between WII and UKB. I also appreciate the response to my comment about the importance of reducing the data down from all combinations of the 4 conceptual dimensions for parsimony.

OUR RESPONSE: We thank the reviewer for these comments.

COMMENT 2. However, I'm still struggling to see the evidence that support the conclusion that RAR and chronotype dimensions alone don't fully capture circadian rhythm patterns – where is the evidence that the clusters identified better explain circadian rhythmicity or the total sleep-wake cycle? Can the authors assess performance of RAR and chronotype metrics alone in explaining circadian patterns compared with the nine derived clusters in the WII and UKB datasets, or demonstrate improved predictive validity of the clusters vs. RAR and chronotype alone for relevant health outcomes and/or behavioural traits? This would be required to justify the conclusions around “Better characterization of circadian rhythm profiles in real-world data allows their impact on adverse health events to be examined with greater accuracy”, which hasn't been directly assessed. While some health-related factors were investigated, these were treated as exposures and it would be of interest to investigate prospective associations with later disease/ill health as the outcome.

OUR RESPONSE: Thank you. As the reviewer knows recent accelerometers provide data on the total sleep-wake cycle, and the commonly used approach takes limited aspects of the data (RAR and chronotype metrics) to draw conclusions on circadian rhythm. Our approach uses all available data encompassing the entire sleep-wake cycle. We understand the reviewer's comment on demonstration of improved predictive ability and we have added new analysis to compare the predictive value for mortality of the commonly used RAR and chronotype metrics alone (10 metrics) with that of all four dimensions (36 metrics). The following changes have been made to the revised manuscript.

Methods section, page 16 paragraph 4:

“To justify use of 36 rather than the 10 commonly used metrics (interdaily stability, intradaily variability, relative amplitude, cosinor mesor, cosinor amplitude, cosinor acrotime, M_{10} and L_5 timing and mean acceleration^{6,8,47-49}) to characterize circadian rhythm we compared the predictive performance of these two sets of metrics for risk of mortality. We first used principal component analysis (PCA) to extract components to reflect the metrics too strongly correlated to be used in the prediction analysis. The number of components for each set of metrics (10 and 36) was selected using the eigenvalue criterion (≥ 1) and the cumulated percentage of variance explained (at least 75%). Cox regression analysis was used to examine association of these components with mortality, and their predictive performances compared using C-index.”

Results section, page 4 paragraph 3:

*“To justify use of 36 rather than commonly used 10 circadian rhythm metrics (including interdaily stability, intradaily variability, relative amplitude, cosinor mesor, cosinor amplitude, cosinor acrotime, M_{10} and L_5 timing and mean acceleration, excluding most daytime activity and sleep metrics) we compared the predictive performance for mortality of the 10 metrics and all 36 metrics. This was done by first using principal components analysis (PCA) in each set of metrics (10 and 36 metrics). A total of three principal components were retained in analysis on 10 metrics (**Table S2-S3**) that explained 80.4% and 77.6% of the variability in WII and UKB datasets, respectively. For 36 metrics, eight principal components were retained that explained 86.5% and 85.4% of the variability in WII and UKB datasets, respectively (**Tables S4-S6**). Over median 11.0 and 8.0 years of follow-up, a total of 632 and 3,001 participants died in WII and UKB, respectively. The predictive performance of the model with 36 metrics (C-index=0.675, 95% confidence interval (CI)=0.664-0.686 in WII; and C-index=0.651, 95%CI=0.640-0.662 in UKB) was higher than the model with 10 metrics (C-index=0.643, 95%CI=0.632-0.654 in WII; and C-index=0.621, 95%CI=0.616-0.627 in UKB), p for difference<0.001 for both cohorts. Subsequent analyses were therefore based on 36 metrics.”*

Discussion section, page 9, paragraph 2:

“Circadian rhythm is altered at older ages² and in individuals with chronic diseases.^{2,3} There is increasing interest in using accelerometer-based measures of RAR and/or chronotype to examine associations of circadian rhythm with adverse health outcomes.^{8,12,14} These studies have generally not considered daytime activity or sleep to measure circadian rhythm, and cannot be seen as a comprehensive reflection of the sleep-wake cycle. Some studies^{6,9,16} have included M₁₀ (mean acceleration during most active 10-hours) and L₅ (mean acceleration during 5 least active hours) as broad indicators of PA and sleep, respectively, but this approach does not fully capture differences in activity intensity (for example LIPA vs MVPA) or sleep disruptions such as fragmentation and number of sleep bouts. Previous studies have shown both sleep^{17,18} and PA¹⁹, two key dimensions of the sleep-wake cycle, to be important for health. Results from our preliminary analysis show use of all 36 metrics to have better predictive performance for mortality than the prediction model with 10 commonly used metrics (RAR, chronotype, M₁₀ and L₅). Furthermore, eight of the nine circadian rhythm profiles we identified were characterized by RAR but also presented differences in the other dimensions. These findings suggest that focusing on RAR and/or chronotype alone may introduce misclassification bias in studies of sleep-wake cycles and health.”

COMMENT 3. Title. I’m not convinced that this is an improvement on the previous one, in particular what is meant by “clusters of circadian rhythm”? Perhaps augment with “population clusters of circadian rhythm profiles” or something similar?

OUR RESPONSE: Apologies, restrictions on word count imply that the title is never quite complete. We have modified the title to “Profiles of circadian rhythm derived from accelerometer-based rest-activity rhythm, chronotype, daytime activity, and sleep in 2 cohort studies”.

COMMENT 4. Abstract. Rephrase to remove double plurals, “Clusters participants differed on sociodemographic, behavioural and health-related factors” and “as demonstrated by the different dimensions combinations identified within clusters”

OUR RESPONSE: Thank you, we have rephrased these sentences as follows:

“The participants within the 9 clusters differed on sociodemographic, behavioural and health-related factors.”

“Previous studies have focussed on RAR alone to measure circadian rhythm and our study suggests that this might be an oversimplification, as demonstrated by nine clusters characterised by specific combinations of RAR, daytime activity, sleep, and chronotype.”

COMMENT 5. In addition, I’m not sure what is meant by – “Focussing only on RAR to measure CR might be an oversimplification, as demonstrated by the different dimensions combinations identified within clusters” ? Can you justify why it is a simplification to only focus on RAR.

OUR RESPONSE: As described in our response to comment 2, it does to be an oversimplification to focus only on one aspect of circadian rhythm. We have now modified this sentence in the abstract so that it is complete.

“Most studies have focussed on RAR alone to measure circadian rhythm and our study suggests that this might be an oversimplification, as demonstrated in the nine clusters characterised by specific combinations of RAR, daytime activity, sleep, and chronotype.”

We have also added the following paragraph in the discussion section to clarify this point. (Page 9, paragraph 2)

“Circadian rhythm is altered at older ages² and in individuals with chronic diseases.^{2,3} There is increasing interest in using accelerometer-based measures of RAR and/or chronotype to examine associations of circadian rhythm with adverse health outcomes.^{8,12,14} These studies have generally not considered daytime

activity or sleep to measure circadian rhythm, and cannot be seen as a comprehensive reflection of the sleep-wake cycle. Some studies^{6,9,16} have included M_{10} (mean acceleration during most active 10-hours) and L_5 (mean acceleration during 5 least active hours) as broad indicators of PA and sleep, respectively, but this approach does not fully capture differences in activity intensity (for example LIPA vs MVPA) or sleep disruptions such as fragmentation and number of sleep bouts. Previous studies have shown both sleep^{17,18} and PA¹⁹, two key dimensions of the sleep-wake cycle, to be important for health. Results from our preliminary analysis show use of all 36 metrics to have better predictive performance for mortality than the prediction model with 10 commonly used metrics (RAR, chronotype, M_{10} and L_5). Furthermore, eight of the nine circadian rhythm profiles we identified were characterized by RAR but also presented differences in the other dimensions. These findings suggest that focusing on RAR and/or chronotype alone may introduce misclassification bias in studies of sleep-wake cycles and health.”

COMMENT 6. Results. Every time a cluster is mentioned, please include in brackets the corresponding descriptives (e.g. RAR--/PA--/chronotype-) to aid interpretation

OUR RESPONSE: Thank you, we have revised as requested by adding the corresponding meaning in brackets to clusters that were referred to in isolation. In other cases, the characteristics of the clusters are included in the text.

COMMENT 7. When describing associations, please state that the clusters are the dependent variables in the Results as well as the Methods.

OUR RESPONSE: Thank you, we have revised as requested.

COMMENT 8. Can p-values be added to Tables 3 and 4?

OUR RESPONSE: Thank you; given the size of the table we have added * to denote $p < 0.05$ and ** to denote $p < 0.01$ for readability.

COMMENT 9. Please give exact p-values in the text rather than “was not statistically significant”

OUR RESPONSE: Thank you, we have revised as requested.

COMMENT 10. Was any consideration given to shift work patterns?

OUR RESPONSE: Unfortunately, these data were not available in the Whitehall II study. In UK Biobank, although available, these data were missing for too many participants (N=20,528). We chose to add the distribution of shift work by clusters in sensitivity analysis (Supplementary Table S10).

Results section, page 7, paragraph 1:

*“Participants in UKB who worked shifts (N/Total N with available data=4,441/34,467) were more likely to be in cluster 1 (RAR++/PA++), 2 (RAR+/PA+/Sleep-), 7 (RAR-/PA-/Sleep--) and 9 (RAR--/PA--/Chronotype-), all $p \leq 0.01$ for comparison with cluster 3 (RAR+/LIPA+/Sleep+), **Table S10.**”*

Methods section, page 16, paragraph 1:

“Data on shift work were only available in UKB and were used in additional analyses, categorized as: “No” and “Yes”.”

COMMENT 11. Discussion. Can some general patterns be identified between the cluster characteristics and sociodemographic/behavioural/health factors in the discussion?

OUR RESPONSE: Thank you, we first report findings from previous studies and then present how our study adds to this previous evidence. This reads as follows:

Discussion section, page 11-12, paragraph 2-3 (11) and 1 (12):

“... in both studies participants’ sociodemographic, behavioural, and health-related factors differed across the clusters. Previous studies have reported participants’ characteristics to vary across metrics such as interdaily stability and intradaily variability,^{16,32,33} relative amplitude, and M₁₀ (most active 10-hour period) and L₅ (least active 5-hour period).¹⁶ Older age has been shown to be associated with a more stable^{32,33} but fragmented RAR^{5,16,32} – as denoted by higher interdaily stability and intradaily variability, – and less daytime activity.¹⁶ Women and people married/cohabiting have been shown to have more stable and less fragmented rhythm.^{32,33} Smoking has also been found to be associated with a less stable rhythm^{16,32,33} while higher BMI was associated with less stable but also more fragmented rhythm.^{16,32}

Our results add to the existing evidence by showing heterogeneity in associations with sociodemographic, behavioural and health-related factors for a given level of RAR depending on the other dimensions of circadian rhythm - daytime activity, sleep, and chronotype. We found women and persons married/cohabiting to have a more stable and less fragmented rhythm, undertake light intensity physical activity, and have good sleep levels. Living in a more deprived area was more common in clusters characterized by sleep disturbances and delayed chronotype in WII, whereas in UKB it was more common in clusters with poorer RAR. Participants with greater outdoor light exposure had more robust RAR and were likely to be active, independently of the season, as previously shown in UKB using self-reported time spent outdoors.³⁴ As the number of blind participants was small (<0.01% in WII and UKB), this group is unlikely to influence findings on light exposure. Participants in clusters characterised by poor sleep and/or disturbed RAR were more likely to be on CNS medications. Poor health status indicated by prevalent diabetes or chronic diseases tended to be more frequent in participants with poorer RAR and less physical activity.”

COMMENT 12. Data on depressive symptoms and functional limitations (assessed through frailty index measures) are available in UK Biobank and so it’s unclear why these have not been included.

OUR RESPONSE: We agree with the reviewer that UK Biobank is a rich dataset. We chose variables for our analyses that were comparable in the two studies. For example, frailty index in UK Biobank includes 49 self-reported conditions and some of these are part of the health-related factors that we used in our analyses. Furthermore, as stated in the introduction our primary aim was to use all four dimensions (RAR, daytime activity, sleep, and chronotype) of the sleep-wake cycle to identify how these measures cluster in older adults. To ensure construct validity of the method, analyses were undertaken in the Whitehall II and UK Biobank (UKB) cohort studies. Only the secondary objective was to determine the sociodemographic, behavioural, and health-related factors correlates of circadian rhythm clusters identified in our study. We chose to focus on a broad set of sociodemographic, behavioural, and health-related factors – the measures chosen are not comprehensive as our primary objective was not to examine all available data in these cohort studies. To avoid confusion, we removed Table S8 on depressive symptoms and functional limitations in Whitehall II from the supplementary analysis. This ensures that similar health-related factors were examined in both studies.

Reviewer #2

The authors have thoroughly addressed all my comments, as well as those raised by others, to the best of my knowledge. With that said, I have no further remarks.

OUR RESPONSE: We thank the reviewer for this feedback, and we appreciate the comment that our revision addressed concerns raised by all reviewers.

Reviewer #3

I am happy with the authors response to my comments.

OUR RESPONSE: Thank you.